# GENERATING GRAPHS VIA SPECTRAL DIFFUSION

**Giorgia Minello**
Ca' Foscari University
giorgia.minello@unive.it

**Alessandro Bicciato**
Ca' Foscari University
alessandro.bicciato@unive.it

**Luca Rossi**
The Hong Kong Polytechnic University
luca.rossi@polyu.edu.hk

**Andrea Torsello**
Ca' Foscari University
andrea.torsello@unive.it

**Luca Cosmo**
Ca' Foscari University
luca.cosmo@unive.it

## ABSTRACT

In this paper, we present GGSD, a novel graph generative model based on 1) the spectral decomposition of the graph Laplacian matrix and 2) a diffusion process. Specifically, we propose to use a denoising model to sample eigenvectors and eigenvalues from which we can reconstruct the graph Laplacian and adjacency matrix. Using the Laplacian spectrum allows us to naturally capture the structural characteristics of the graph and work directly in the node space while avoiding the quadratic complexity bottleneck that limits the applicability of other diffusion-based methods. This, in turn, is accomplished by truncating the spectrum, which, as we show in our experiments, results in a faster yet accurate generative process, and by designing a novel transformer-based architecture linear in the number of nodes. Our permutation invariant model can also handle node features by concatenating them to the eigenvectors of each node. An extensive set of experiments on both synthetic and real-world graphs demonstrates the strengths of our model against state-of-the-art alternatives.

## 1 INTRODUCTION

Generating realistic graphs by learning from a distribution of real-world graphs has gained increasing attention from researchers in many fields due to its wide range of applications. For instance, synthetic graph generation plays a crucial role in drug design Gómez-Bombarelli et al. (2018); Li et al. (2018a); You et al. (2018a) as well as in network science Watts & Strogatz (1998); Leskovec et al. (2010); Albert & Barabási (2002).

Seminal graph generation approaches date back to the 1960s and rely on simple stochastic processes, limiting their ability to capture complex dependencies seen in real-world networks. For example, the Barabási-Albert Albert & Barabási (2002) and Kronecker Leskovec et al. (2010) graph models are specifically designed to generate graphs belonging to specific families and lack the ability to learn directly from observed data. While these models may excel in capturing a set of predefined properties, they are often unable to represent a wider range of aspects observed in real-world graphs. In addition, in several domains, network properties are largely unknown, which further limits the applicability of these techniques. For instance, the Barabási-Albert model Albert & Barabási (2002) allows to create graphs that exhibit the scale-free nature found in empirical degree distributions, however it is unable to capture other facets of real-world graphs, *e.g.*, community structure. While a flurry of new models attempting to address these shortcomings have been introduced by the network science community (see Drobyshevskiy & Turdakov (2019) for a recent review), these methods often lack the ability to learn to mimic the characteristics of a given dataset. This in turn limits the expressivity and fidelity of generated graphs and thus the range of possible applications of graph generative models.

In this paper, we introduce a new model for Generating Graphs via Spectral Diffusion (GGSD). The ideas underpinning our approach are 1) to represent the graph using the eigendecomposition of its Laplacian matrix and 2) to use a diffusion-based approach to learn to sample sets of eigenvalues and eigenvectors from which a graph adjacency matrix can be reconstructed. Doing this allows us to work directly in the space of nodes while overcoming the computational bottleneck (quadratic in the number of graph nodes) of other methods that follow a similar approach Vignac et al. (2022). By limiting the number of eigenvalues and eigenvectors used to reconstruct the graph adjacency matrix, we reduce the complexity of the iterative denoising process to be linear with respect to the number of nodes while, at the same time, having a representation tablet to encapsulate graph structural characteristics. Moreover, unlike other models conditioned on spectral representations Martinkus et al. (2022), our model also allows us to robustly condition the generation of new graphs on desired spectral properties (subsets of eigenvalues and/or eigenvectors) at inference time.

The remainder of this paper is structured as follows. Section 2 reviews the related work, while Sections 3 introduces the necessary background on denoising diffusion models. We introduce our graph generative model in Section 4, and we present the experimental evaluation against state-of-the-art alternatives in Section 5. Finally, Section 6 concludes the paper.

## 2 RELATED WORK

In contrast to the image and text domains, where the development of generative models is well understood and established, graphs introduce a series of additional challenges.

The first issue is the non-uniqueness of graph representations, *i.e.*, if a graph contains $n$ nodes, there exist up to $n!$ possibly distinct adjacency matrices that serve as equivalent representations of the same graph, since there is no reason to prefer a particular node order. Ideally, a generative model should assign equal probability to each of these $n!$ adjacency matrices. Another crux lies in the size of the output space, which is quadratic in the number of nodes, and that quickly becomes a bottleneck when dealing with large graphs. Graph generative models should also be able to consider the existence of dependencies and relationships between nodes and edges, rather than treating them as independent, *e.g.*, in social networks the likelihood of two nodes being connected is often higher when they have common neighbors. Finally, standard machine learning techniques designed for continuously differentiable objective functions are unsuitable to be directly applied to discrete graph structures Guo & Zhao (2022).

Seminal graph generative model approaches seek to address these problems, yet focus only on the generation of graphs displaying a limited set of structural characteristics. These initial methods rely on identifying common characteristics in real-world graphs, such as degree distribution, graph diameter, and clustering coefficient Faloutsos (2008), and then generate synthetic graphs through the application of a set of heuristic rules Leskovec et al. (2010); Leskovec & Faloutsos (2007); Erdős et al. (1960); Albert & Barabási (2002). Although these models can produce synthetic graphs with the given desired features, they are limited in their ability to generate node features as well as novel structural patterns.

A breakthrough in this field has been marked by the recent progress in deep learning models such as Variational Auto Encoders (VAEs) Kingma & Welling (2013), Recurrent Neural Networks (RNNs) Zaremba et al. (2014) and Generative Adversarial Networks (GANs) Goodfellow et al. (2014). In this context, we encounter models commonly referred to as *general-purpose deep graph generative models*, such as GraphRNN You et al. (2018b) and GRAN Liao et al. (2019), which exploit deep architectures to learn the graphs distribution. Even though they represent a step forward in the field of generative graph models, most of them are limited by exclusively focusing on the graphs structure. Further, approaches of this type adopt evaluation metrics based only on graph statistics, like degree distribution or clustering coefficients, and thus overlook or omit the assessment of the generated node features.

Node and edge features are instead considered in a number of methods developed specifically for the generation of molecules, indeed one of the most promising application scenario for modern graph generation approaches. Models falling in this domain, referred to as *molecule graph generative models*, exploit deep architectures such as GAN in De Cao & Kipf (2018) or RNN in Popova et al. (2019) as well as other generation strategies (*e.g.*, graph normalizing flows Luo et al. (2021)) or the

combination of different approaches. For instance, Shi et al. (2020) combines the advantages of both autoregressive and flow-based methods.

Nevertheless, there are other deep learning approaches beyond molecule graph generative models that are capable of generating graphs with node and edge features - even though the evaluation itself is often still based on a molecule generation task. For instance Simonovsky & Komodakis (2018) and Grover et al. (2019) propose general deep generative models for graphs based on variational autoencoders. The main drawback of these architectures is that they are specialized and limited to small-scale graphs with low-dimensional feature space Yoon et al. (2023).

Another category of graph generative models takes cues from the score-based generative modeling work of Song & Ermon (2019) to define diffusion models for graphs. For instance, in Huang et al. (2022) the authors propose a forward diffusion process, specifically a continuous-time generative diffusion process for permutation invariant graph generation. Similarly, Niu et al. (2020) introduce a different diffusion model named Edge-wise Dense Prediction Graph Neural Network (EDP-GNN), which uses Gaussian noise and uses thresholding to address the issue of generating a discrete valued adjacency matrix from continuous values. Crucially, the proposed method cannot fully capture node-edge dependencies. A similar score-based generative model for graphs, where both node features and adjacency matrix are created, is presented in Jo et al. (2022). Finally, Vignac et al. (2022) suggest an alternative approach where a discrete diffusion process is used to generate graphs with discrete node and edge features. This is similar to Haefeli et al. (2022), however the latter can only be applied to unattributed graphs.

More recently, Martinkus et al. (2022) with their SPECTRE network and Luo et al. (2023) take a different approach by considering the graph spectra, thus leveraging the inherent ability of the low frequency portion of the spectrum to capture global structural characteristics of the corresponding graph. Although similar to our method, SPECTRE focuses on generating an adjacency matrix conditioned on a set of eigenvectors, which may or may not have been generated themselves. Our method instead only generates eigenpairs from which the adjacency matrix is recovered. As a result, unlike SPECTRE, our method is capable of generating graphs that respect a set of given spectral properties (see Subsection 5.4). GSDM Luo et al. (2023), on the other hand, proposes to reduce the complexity by performing diffusion just on the eigenvalues and optionally on the node features, while the eigenvectors used to reconstruct the final adjacency matrix are uniformly sampled from the training set. DiGress Vignac et al. (2022) is also closely related to our model, however its complexity is quadratic in the number of nodes of the graph, making it unsuitable to work on large graphs.

Our approach is also related to recently introduced latent graph diffusion models, which employ an autoencoder architecture to map nodes and edges of a graph to latent space where the diffusion process takes place Yang et al. (2024); Zhou et al. (2024). While these approaches aim to learn a low-dimensional embedding of the graph nodes, we rely instead on the well-established concept of spectral embedding Luo et al. (2003), with eigenvectors providing a low-dimensional embedding of the graph nodes and eigenvalues capturing global structure information.

## 3   DENOISING DIFFUSION MODELS

Denoising Diffusion Probabilistic Models (DDPMs) are a class of generative models inspired by considerations from non-equilibrium thermodynamics. In particular, diffusion models in deep learning were first introduced in Sohl-Dickstein et al. (2015) yet popularized only in 2020 Ho et al. (2020). They operate by iteratively introducing noise to an input signal and then learning to denoise it thus generating new samples from the corrupted signals. Specifically, the idea is to destroy the structure in a data distribution through an iterative forward diffusion process (noising) and then learn a reverse diffusion process (denoising). This reverse process restores structure in the data, thus yielding a tractable generative model of the data.

Given a data point sampled from a real but unknown data distribution $\mathbf{x_0} \sim q(\mathbf{x})$, we define a forward noising process $q$ producing a sequence of noisy samples $\mathbf{x}_1, \ldots, \mathbf{x}_T$ as a Markov Chain given by $q(\mathbf{x}_1, \ldots, \mathbf{x}_T \mid \mathbf{x}_0) = \prod_{t=1}^{T} q(\mathbf{x}_t \mid \mathbf{x}_{t-1})$, with the diffusion kernel defined as:

$$q(\mathbf{x}_t \mid \mathbf{x}_{t-1}) = \mathcal{N}\left(\mathbf{x}_t; \sqrt{1 - \beta_t}\mathbf{x}_{t-1}, \beta_t I\right). \tag{1}$$

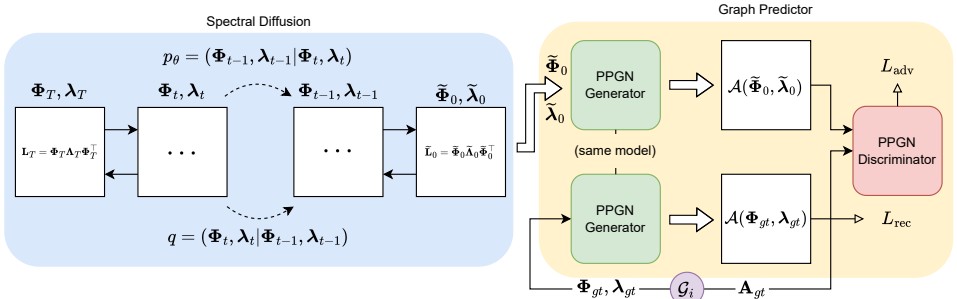

Figure 1: GGSD pipeline. During the spectral diffusion process (left) the neural network is trained to predict the denoising steps for the eigenvectors $\phi$ and eigenvalues $\lambda$ of the graph Laplacian. The second stage of our method is the graph predictor (right), where we train a Provably Powerful Graph Network (PPGN) Maron et al. (2019) (similar to what was done in SPECTRE Martinkus et al. (2022)). Given the eigenvalues and eigenvectors generated, it predicts the adjacency matrix.

Note that, if we define $\bar{\alpha}_t = \prod_{s=1}^t (1 - \beta_s)$, we can reformulate Eq. 1 as a single step

$$q(\mathbf{x}_t \mid \mathbf{x}_0) = \mathcal{N}\left(\mathbf{x}_t; \sqrt{\bar{\alpha}_t}\mathbf{x}_0, (1 - \bar{\alpha}_t)\mathbf{I}\right) . \qquad (2)$$

In the reverse diffusion process, the goal is to recreate the true sample from a Gaussian noise input $\mathbf{x}_T \sim \mathcal{N}(\mathbf{0}, \mathbf{I})$ by sampling from $q(\mathbf{x}_{t-1} \mid \mathbf{x}_t)$, the true denoising distribution. In order to run the reverse diffusion process, we need to learn a model $p_\theta$, often referred to as *score model*, to approximate these conditional probabilities. As Feller (1949) showed, in the case of Gaussian distributions the diffusion process reversal has the same functional form of the forward process. From this it follows that the reverse diffusion process kernel can be defined as

$$p_\theta(\mathbf{x}_{t-1} \mid \mathbf{x}_t) = \mathcal{N}(\mathbf{x}_{t-1}; \boldsymbol{\mu}_\theta(\mathbf{x}_t, t), \boldsymbol{\Sigma}_\theta(\mathbf{x}_t, t)), \qquad (3)$$

where $\theta$ are the parameters of the reverse diffusion kernel at each time step, which can be learned using a neural network. If we fix the variance to a constant $\beta_t$ (*i.e.* $\boldsymbol{\Sigma}_\theta(\mathbf{x}_t, t) = \beta_t I$), we only need to learn the distance between the means of two Gaussian distributions, *i.e.*, between the noise added in the forward process and the noise predicted by the model. This leads to a variational lower bound loss expressed in terms of the Kullback–Leibler (KL) divergence between the posterior of the forward process and the parameterized reverse diffusion process.

## 4 OUR METHOD

Consider an undirected unweighted graph $\mathcal{G} = (\mathcal{V}, \mathcal{E})$, where $\mathcal{V}$ is the set of $n$ nodes connected by the edge set $\mathcal{E}$. Recall that for a graph with adjacency matrix $\mathbf{A}$, the graph Laplacian $\mathbf{L}$ is defined as $\mathbf{L} = \mathbf{D} - \mathbf{A}$, where $\mathbf{D}$ is the diagonal degree matrix. Finally, let $\boldsymbol{\Phi}$ and $\boldsymbol{\Lambda}$ be the orthonormal matrix of eigenvectors (as columns) and the diagonal matrix of eigenvalues given by the eigendecomposition $\mathbf{L} = \boldsymbol{\Phi}\boldsymbol{\Lambda}\boldsymbol{\Phi}^\top$, respectively. In the following sections, we use $\boldsymbol{\lambda}$ to denote the vector of eigenvalues of the graph Laplacian.

The fundamental intuition underpinning our model is that we can represent the graph connectivity with (possibly a subset of) the eigenvectors $\boldsymbol{\Phi}$ and the corresponding eigenvalues $\boldsymbol{\lambda}$ of the graph Laplacian. The connection between the spectrum of the graph Laplacian and the structural properties of the underlying graph is well known and studied. For example, it is well established that the low frequency portion of the spectrum captures the global structural characteristic of the graph, while the high frequencies are essential in the reconstruction of local connectivity patterns Chung (1997).

Fig. 1 shows an overview of the proposed pipeline. This is made of two main components, namely 1) a spectral diffusion process that generates a set of eigenvalues and eigenvectors from which an approximation of the Laplacian matrix can be reconstructed and 2) a graph predictor, which outputs a binary adjacency matrix from the (noisy) Laplacian reconstruction. The two components are discussed in detail in the following subsections.

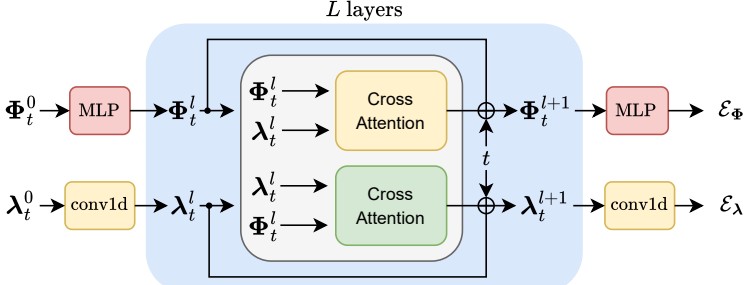

Figure 2: The score model takes as input the noisy eigenvector matrix and eigenvalues at time $t$ and predicts the noise of the data to be used in the denoising step. The $k$ node feature eigenvectors $\mathbf{\Phi}_t^0$ are projected through an MLP to a $d$ dimensional space. The sequence of $k$ eigenvalues is given as input to a 1D convolutional layer, which outputs $d$ features for each eigenvalue. Both eigenvectors and eigenvalues go through a series of $L$ layers composed of two multi-head cross-attention blocks, one updating the eigenvectors conditioned by the eigenvalues and one updating the eigenvalues conditioned on the eigenvectors. After each layer, we apply a residual block $\oplus$, which adds to the layer input the updated values scaled and shifted by time-dependent factors. Finally, $\mathbf{\Phi}_t^L$ and $\boldsymbol{\lambda}_t^L$ are projected to a $k$ dimensional space through an MLP and a 1D convolution.

## 4.1 SPECTRAL DIFFUSION

Moving from the Laplacian to its spectrum reduces the double row-, column-covariance with respect to node permutations of the Laplacian matrix, to a single covariance over the rows of the eigenvector matrix. To address this covariance, we represent the eigenvector matrix as a series of spectral embedding of the nodes, *i.e.*, we interpret the $i$-th component of the $j$-th eigenvector as the $j$-th component of the $i$-th node embedding, or alternatively, we see the rows of $\mathbf{\Phi}$ as vectors. To this we add the eigenvalues $\boldsymbol{\lambda}$ as a global graph descriptor. We can then define the reverse diffusion step as

$$p_\theta(\mathbf{\Phi}_{t-1}, \boldsymbol{\lambda}_{t-1} | \mathbf{\Phi}_t, \boldsymbol{\lambda}_t) = \mathcal{N}\left(\{\mathbf{\Phi}_{t-1}, \boldsymbol{\lambda}_{t-1}\}; \boldsymbol{\mu}_\theta\left(\mathbf{\Phi}_t, \boldsymbol{\lambda}_t, t\right), \sigma_t^2 \mathbf{I}\right) \quad (4)$$

where the normal distribution is over the product set of the spectral embeddings and the global spectral descriptor.

Following DDPM Ho et al. (2020), we train a neural network to predict the denoising step. We design the backbone of our spectral diffusion process of Eq. 4 as a neural network composed of a sequence of layers containing a pair of multi-head attention blocks Vaswani et al. (2017), one operating on the eigenvectors conditioned on the eigenvalues and one operating on the eigenvalues conditioned by the eigenvectors. This choice allows us to achieve both a node permutation invariant model and to let the eigenvectors and eigenvalues condition each other on the prediction of the denoising step. Moreover, this model can easily handle node features $\mathbf{X}$ by simply concatenating them to the eigenvectors of each node. Fig. 2 shows the overall structure of the proposed neural network. Note that the `conv1d` layer is applied to the eigenvalues, which are invariant with respect to node permutations, while the diffusion process itself acts in an invariant way on node embeddings, which are permutationally covariant. As a consequence, the whole process is permutationally invariant.

Crucially, our model allows us to reduce the memory footprint of the diffusion component from $O(n^2)$ to $O(kn)$, where $n$ denotes the number of graph nodes, by fixing the maximum number of eigenvectors to $k$, resulting in a faster generative process. For this reason, for larger graphs, we perform diffusion on a subset of $k$ eigenvectors. In this case, after the denoising diffusion process, we obtain a subset $\tilde{\mathbf{\Phi}}$ of columns of $\mathbf{\Phi}$ with the corresponding subset of eigenvalues $\tilde{\boldsymbol{\lambda}}$, which allows for an approximated reconstruction of $\tilde{\mathbf{L}} = \tilde{\mathbf{\Phi}} \tilde{\mathbf{\Lambda}} \tilde{\mathbf{\Phi}} \approx \mathbf{L}$, from which the adjacency matrix can be inferred. In addition, since the eigenvector associated with the null eigenvalue does not contribute to the reconstruction of the Laplacian matrix, we can safely ignore it in the generation process.

Note that the optimal number of eigenvalues/vectors ($k$) to use is not fixed. We determined its range through preliminary analyses on the SBM and Planar datasets (see Appendix E), and in general, we treat it as a hyper-parameter of the model. Interestingly, our experiments reveal that the eigenvectors corresponding to the smallest eigenvalues (lowest frequencies) do not consistently offer more information about connectivity or result in better reconstructions of the original adjacency matrix.

Table 1: Comparison with other graph generative models using MMD metrics (the smaller, the better) on synthetic datasets.

| | Community-Small | | | | Planar | | | | Stochastic Block Model (SBM) | | | |
|---|---|---|---|---|---|---|---|---|---|---|---|---|
| | Deg. ↓ | Clus. ↓ | Spect. ↓ | Orb. ↓ | Deg. ↓ | Clus. ↓ | Spect. ↓ | Orb. ↓ | Deg. ↓ | Clus. ↓ | Spect. ↓ | Orb. ↓ |
| GraphRNN | 0.0271 | 0.1072 | 0.0520 | 0.1469 | 0.0096 | 0.2985 | 0.0389 | 1.4022 | 0.0178 | 0.0151 | 0.0104 | 0.0351 |
| GRAN | **0.0013** | 0.0843 | 0.0282 | 0.0201 | 0.0202 | 0.2985 | 0.0248 | 0.1964 | 0.0135 | 0.0149 | 0.0034 | 0.0352 |
| DiGress | 0.0096 | 0.1035 | 0.0506 | 0.0372 | **0.0005** | **0.0178** | **0.0020** | 0.0115 | 0.0166 | 0.0246 | 0.0064 | 0.1327 |
| GSDM | 0.0099 | **0.0446** | **0.0131** | 0.0155 | 0.0220 | 0.0222 | 0.0096 | 0.0371 | 0.2295 | 0.2280 | 0.1578 | 0.2876 |
| GDSS | 0.0107 | 0.1060 | 0.0450 | 0.0356 | 0.0701 | 0.3025 | 0.0403 | 1.0345 | 0.2658 | 0.0442 | 0.0551 | 0.2780 |
| SPECTRE | 0.0079 | 0.1067 | 0.0460 | 0.0250 | 0.0008 | 0.0859 | 0.0147 | 0.0058 | 0.0044 | 0.0118 | **0.0015** | **0.0140** |
| GGSD | 0.0016 | 0.0590 | 0.0153 | **0.0142** | 0.0007 | 0.1881 | 0.0125 | **0.0047** | **0.0005** | **0.0115** | 0.0045 | 0.0289 |

## 4.2 GRAPH PREDICTOR

The main drawback of considering a subset of the eigenvectors is the introduction of noise on the reconstructed adjacency matrix. We adopt a strategy similar to the one proposed in SPECTRE Martinkus et al. (2022) to predict a binary adjacency matrix starting from a noisy reconstruction. We train a *l*-layer Provably Powerful Graph Network (PPGN) Maron et al. (2019), which takes as input the generated eigenvectors $\tilde{\mathbf{\Phi}}$ scaled by the square root of the eigenvalues $\tilde{\mathbf{\lambda}}$ as node features as well as the noisy adjacency matrix $\tilde{\mathbf{A}} = \tilde{\mathbf{D}} - \tilde{\mathbf{L}}$ and predicts the binary adjacency matrix

$$\mathcal{A}(\tilde{\mathbf{\Phi}}, \tilde{\mathbf{\lambda}}) = \sigma(\text{PPGN}_l(\tilde{\mathbf{A}}, \tilde{\mathbf{\Phi}}\tilde{\mathbf{\Lambda}}^{\frac{1}{2}})), \tag{5}$$

where $\sigma$ is a sigmoid activation function and $\text{PPGN}_l$ is a sequence of PPGN layers. We train this network with two losses: 1) a reconstruction loss encouraging the adjacency matrix predicted from the reduced ground-truth eigenvectors $\tilde{\mathbf{\Phi}}gt$ and eigenvalues $\tilde{\mathbf{\lambda}}_{gt}$ to match the corresponding input adjacency matrix $\mathbf{A}_{gt}$, and 2) an adversarial loss on the generated adjacency matrix $\mathcal{A}(\tilde{\mathbf{\Phi}}, \tilde{\mathbf{\lambda}})$, *i.e.*,

$$L_{\text{rec}} = \text{BCE}(\mathbf{A}_{gt}, \mathcal{A}(\tilde{\mathbf{\Phi}}_{gt}, \tilde{\mathbf{\lambda}}_{gt})) \text{ and } L_{\text{adv}} = \log(\mathcal{D}(\mathbf{A}_{gt})) + \log(1 - \mathcal{D}(\mathcal{A}(\tilde{\mathbf{\Phi}}, \tilde{\mathbf{\lambda}}))), \tag{6}$$

where BCE is the standard binary cross entropy loss and $\mathcal{D}$ is a discriminator network composed of a sequence of PPGN layers followed by a global pooling for the final graph-level classification. Note that, unlike in SPECTRE Martinkus et al. (2022), this refining step is not generative, meaning that the output is deterministic and depends solely on the input eigenvectors/values.

## 5 EXPERIMENTAL EVALUATION

**Datasets.** We compare the performance of our model against that of state-of-the-art alternatives on both synthetic and real-world datasets. In line withh prior work, we use three synthetic datasets and two real-world datasets. The synthetic datasets we consider are (i) Community-small ($12 \leq |V| \leq 20$), (ii) Planar ($|V| = 64$), and (iii) Stochastic Block Model (SBM) (2-5 communities and 20-40 nodes per community). The real-world datasets are both from the molecular domain, namely (i) Proteins (100-500 nodes) Dobson & Doig (2003) and (ii) QM9 (9 nodes) Ruddigkeit et al. (2012); Ramakrishnan et al. (2014). Detailed descriptions of all datasets can be found in Appendix A.

**Evaluation Metrics.** We assess the ability of the models to generate graphs with structural characteristics close to those of the training graphs by following the methodology outlined in Liao et al. (2019), which aims to address the difficulties of measuring likelihoods when evaluating autoregressive graph generative models reliant on orderings. In particular, we adopt the approach proposed by You et al. (2018b) and Li et al. (2018b) and used by many others Krawczuk et al. (2020) as well. Our evaluation centers on contrasting the distributions of graph statistics between the generated and actual graphs. Specifically, we consider the following key graph statistics: 1) degree distribution (Deg.), 2) clustering coefficient (Clus.), 3) eigenvalues of the normalized graph Laplacian (Spec.), and 4) the occurrence frequency of all 4-node orbits (Orb.). Moreover, for QM9, we follow the literature and evaluate the quality of the generated graphs by computing the validity of the generated molecules, their uniqueness, and their novelty w.r.t. to the molecules in the training set and viceversa Samanta et al. (2020); Guo & Zhao (2022). Detailed information on the evaluation metrics used can be found in Appendix B.

**Baselines.** We evaluate the effectiveness of our model by comparing it against a number of well-known graph generative models as well as some recently developed deep graph generative models. In particular, we consider GraphRNN You et al. (2018b), GRAN Liao et al. (2019), DiGress Vignac et al. (2022), SPECTRE Martinkus et al. (2022), GDSS Jo et al. (2022) and GSDM Luo et al. (2023). For the molecule generation task we also include GraphVAE Simonovsky & Komodakis (2018).

**Experimental Setup.** To maximize the robustness of the experimental results, we follow a slightly different experimental setup compared to previous works. Specifically, for the synthetic datasets, we decided to create a larger set of test graphs: 200 graphs for Planar and SBM, and 100 graphs for community-small. Accordingly, we let each model generate an equivalent number of graphs (200 for Planar and SBM, 100 for community-small) to compute the MMD measures. Due to the limited number of graphs in the Proteins dataset (see Appendix A), we also followed a different and more robust protocol to evaluate the generated graphs on this dataset. Rather than utilizing a single subset of the dataset as a test set, we created 10 folds (identical for each method) allowing us to report the average of each metric ($\pm$ standard deviation) over the 10 folds. Further detailed information on the model settings and training setup, both for our model and the baselines, is provided in Appendix D.

## 5.1 EVALUATING THE GENERATED GRAPHS

**Synthetic Datasets** The experimental results for community-small, SBM, and Planar are shown in Table 1. In this table, we report the MMD metrics for the graph statistics, where the smaller the statistics, the better. The results of our method (GGSD) are chosen from those obtained using either the lower or upper range of eigenvalues. Specifically, for the Planar, the results refer to the 16 smallest eigenvalues, whereas for the community-small dataset and SBM we used the largest ones, 8 and 32 respectively. The best performance is highlighted in bold, while the second-best value is underlined. Overall, our model consistently achieves the best or second-best results across all datasets, with the exception of *Clus.* in Planar and *Spect.* in SBM and Planar. We posit that the lower performance on planar graphs may be related to the behaviour of the eigenvectors of this type of graphs. Note in fact that there is no clear class structure in this dataset but rather the graphs are related by the (hard) property of planarity. Indeed, graphs with similar spectra can lie on opposite sides of the discrimination boundary, *i.e.*, between planar and non-planar graphs. As such, the addition or removal of an edge connecting local substructures can easily break the planarity of the graph without significantly affecting its spectral representation. Finally, for Planar and SBM, we also evaluated the quality of the generated graphs in terms of validity, uniqueness, and novelty in Appendix C, showing the ability of our method to generate graphs that are at the same time valid, unique, and novel.

**Real-world Datasets** Results for real-world dataset generations on Proteins and QM9 are reported in Table 2, left and right, respectively. Also in this series of experiments, we achieve good performance. Again, we selected the best results from either the highest or lowest frequencies. Notably, for the Proteins dataset, we utilized the 16 smallest eigenvalues, while for QM9, we used the entire spectrum. The Proteins dataset is especially challenging due to the size of the graphs, which can reach 500 nodes. For this dataset, GGSD ranks as the best in all metrics Table 2 (left). QM9, on the other hand, is composed by small graphs of up to 9 nodes, with both node and edge features. In Table 2 (right), it is important to note that the last column is of particular interest as it summarizes all values, where we emerge as the second top performer.

## 5.2 GRAPH PREDICTOR ABLATION

Given that the "Graph Predictor" is trained using a discriminative loss, one may think that it is this component that is doing all the "heavy lifting" of the graph generation task, while the "Spectral Diffusion" may be just producing noisy data.

To assess that this is not the case, we designed an experiment training our model to predict all the eigenvectors and eigenvalues on the community-small dataset. The small size of the graphs in this datasets allows us to generate all the eigenvectors $\mathbf{\Phi}$ and eigenvalues $\boldsymbol{\lambda}$ and reconstruct the exact Laplacian $\mathbf{L} = \mathbf{\Phi}\mathbf{\Lambda}\mathbf{\Phi}^\top$ and the (almost) binary adjacency matrix $\mathbf{A} = \mathbf{D} - \mathbf{L}$, where $\mathbf{D}$ is the diagonal of the Laplacian. To obtain a binary adjacency matrix, we further threshold $\mathbf{A}$ to get the actual edges of the generated graph (*i.e.*, every entry above $0.5$ is considered an edge).

Table 2: Comparison on real datasets with other graph generative models. **Left**: Proteins, using MMD metrics (the lower, the better) and mean $\pm$ standard deviation (over 10 folds). **Right:** QM9, based on (V)validity, (U)niqueness, and (N)ovelty metrics (the higher, the better). Results denoted by $*$ and $\dagger$ are taken from Martinkus et al. (2022) and Vignac et al. (2022) respectively.

| | Proteins | | | | | QM9 | | |
|---|---|---|---|---|---|---|---|---|
| | Deg. ↓ | Clus. ↓ | Spect ↓ | Orbit ↓ | | Val.↑ | V.& U.↑ | V.& U.& N.↑ |
| GraphRNN | 0.0065±0.0011 | 0.1658±0.0088 | 0.0170±0.0009 | 0.8142±0.0273 | GraphVAE$^*$ | 0.5570 | 0.4200 | 0.2610 |
| GRAN | 0.0569±0.0056 | 0.1622±0.0092 | 0.0146±0.0007 | 0.3430±0.0363 | DiGress$^\dagger$ | 0.9900 | 0.9523 | 0.3180 |
| GSDM | 0.3792±0.0041 | 0.4653±0.0089 | 0.3024±0.0032 | 0.9589±0.0285 | GDSS | 0.8335 | 0.8281 | 0.7257 |
| GDSS | 0.0653±0.0063 | 0.4160±0.0089 | 0.0706±0.0021 | 0.8168±0.0118 | SPECTRE$^*$ | 0.8730 | 0.3120 | 0.2910 |
| SPECTRE | 0.0082±0.0021 | 0.0988±0.0071 | 0.0066±0.0004 | 0.0328±0.0039 | | | | |
| GGSD | **0.0014±0.0003** | **0.0856±0.0066** | **0.0059±0.0007** | **0.0296±0.0066** | GGSD | 0.966 | 0.864 | **0.847** |

In Table 3 we show three different configurations of our method: 1) "Only Diffusion": we use the technique we just described, in which the graph is constructed directly from the generated $\Phi$ and $\lambda$ without using the "Graph Predictor"; 2) "Noise + Predictor": we give as input to the "Graph Predictor" noise drawn from a Gaussian distribution; 3) "Diff. + Prediction": this is the full model used in all the other experiments. For each configuration, we provide results for two cases: 1) using the full set of eigenpairs - all eigenbasis to train the model, and 2) using a subset of the eigenpairs - model trained using the 8 largest eigenvalues and their corresponding eigenvectors.

Reconstructing the graph directly from the Laplacian (referred to as "Only Diffusion") using the complete spectrum yields the best results. Conversely, using either random noise or the reconstructed Laplacian as input to the Predictor results in significantly inferior outcomes. This suggests that the "Spectral Diffusion" part of the network is responsible for the actual generation process. On the other hand, if we consider a truncated eigenbase for the training phase, the Predictor becomes useful as it helps to refine the results further.

To provide a comprehensive overview, we also provide some qualitative examples in Figure 3, comparing the eigenvectors generated by the diffusion module of GGSD with those of the Laplacian computed on the adjacency matrix predicted by the PPGN module. In smaller graphs, the eigenvectors are almost perfectly preserved, while only minor local differences emerge in larger graphs. In contrast, the generative approach used by SPECTRE fails to maintain the relationship between the conditioning eigenvectors and the final generated graph.

## 5.3 EIGENVECTORS ORTHOGONALITY

The general framework of DDPM cannot guarantee the orthonormality of the generated eigenvectors. While in principle this might pose a problem, in practice we observed that this property is well preserved in the generations. In Appendix F, we present both quantitative and qualitative analyses to evaluate the orthogonality of the generated eigenvectors, demonstrating that they exhibit approximate orthonormality. Moreover, in order to test if having an exact orthonormality of the eigenvectors brings any advantage, we tried to reproject the final generated eigenvectors to an orthonormal matrix through QR decomposition before the PPGN predictor step. As reported in Figure 4, we did not observe a clear benefit in having orthonormal basis. We argue that, even if orthonormality is a well-known characteristic of the eigendecomposition of the graph Laplacian (indeed, of the eigendecomposition of any symmetric matrix), it is probably not the most important (nor essential) to guarantee a good reconstruction of a valid graph Laplacian, which exhibits more complex properties that need to be learned directly from data.

Table 3: Ablation of the Graph Predictor network on the community-small dataset.

| | Full Set of Eigenpairs | | | | First 8 Eigenpairs | | | |
|---|---|---|---|---|---|---|---|---|
| | Deg. ↓ | Clus. ↓ | Spect. ↓ | Orb. ↓ | Deg. ↓ | Clus. ↓ | Spect. ↓ | Orb. ↓ |
| Only Diffusion | **0.00134** | **0.05484** | **0.01574** | **0.00514** | 0.00936 | 0.09753 | 0.04341 | 0.01255 |
| Noise + Predictor | 0.04428 | 0.11129 | 0.05691 | 0.46051 | 0.06681 | 0.12330 | 0.05972 | 0.45137 |
| Diff. + Predictor | 0.00562 | 0.07887 | 0.02232 | 0.00942 | **0.00423** | **0.05464** | **0.01832** | **0.00866** |

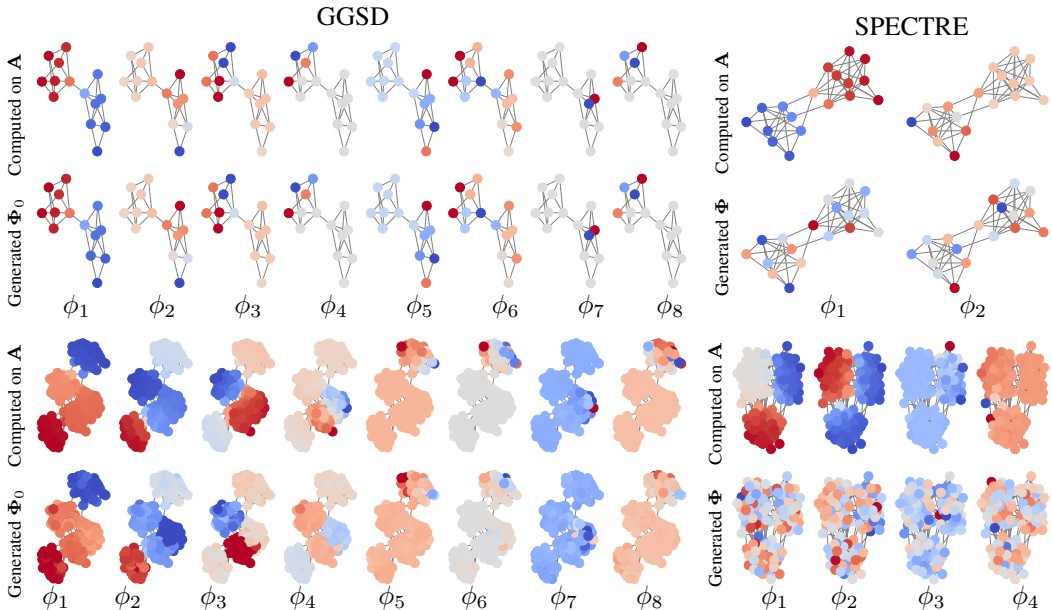

Figure 3: **Left column**: Comparison of the eigenvectors generated by the diffusion module of GGSD with the eigenvectors recomputed on the Laplacian computed on the adjacency matrix predicted by the PPGN module on the Community (top) and SBM (bottom) datasets. The models have been trained with the 8 smallest eigenvectors. In the smaller dataset (Community) the diffusion generates nearly perfect eigenvectors. In the more challenging SBM dataset we can notice that the generated eigenfunction are slightly different from the one computed on the predicted graph while preserving the overall structure. **Right column**: comparison of the interpolated eigenvectors that SPECTRE uses to condition the PPGN module to the actual eigenvectors of the generated graph. In this case the eigenvectors structure is completely lost in the generative process.

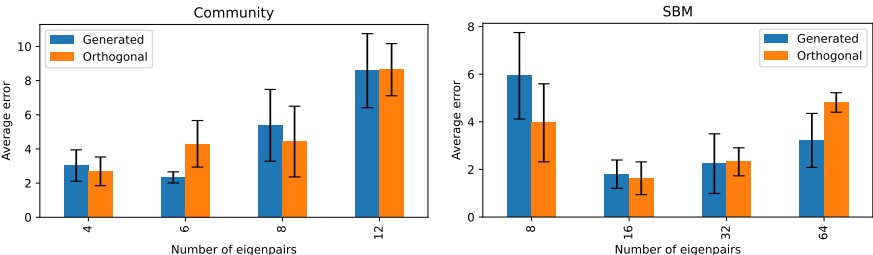

Figure 4: Performance analysis without (Generated) and with (Orthonormal) reprojecting the generated eigenvectors to an orthonormal basis. The average error represents the mean degradation of metrics between the generated graphs and the training set. We report both the mean and the standard deviation as error bars on 10 generations of 200 graphs. Specifically, Degree, Cluster, and Spectral metrics are calculated between the generated graphs and the test set, then normalized by the metrics between the training and test sets.

## 5.4 SPECTRAL CONDITIONED GENERATION

The spectrum of the Laplacian plays an important role in many applications, from graph classification and mesh analysis Cosmo et al. (2020; 2022); Bai et al. (2015); Hu et al. (2014); Minello et al. (2019); Gasparetto et al. (2015a;b) to reconstructing the underlying geometry of a triangulated 3D shape Cosmo et al. (2019); Marin et al. (2021); Moschella et al. (2022) and to define universal adversarial attacks Rampini et al. (2021). Being able to generate a graph given a spectrum is thus an important feature of a generative method. We pose the graph generation problem conditioned on a sequence of eigenvalues and/or eigenvectors as an inpainting problem Lugmayr et al. (2022).

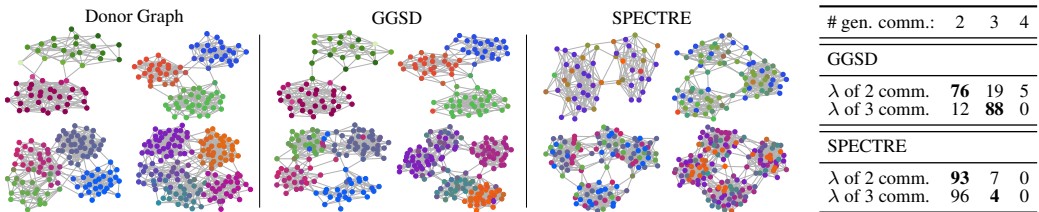

| # gen. comm.: | 2 | 3 | 4 |
|---|---|---|---|
| **GGSD** | | | |
| $\lambda$ of 2 comm. | **76** | 19 | 5 |
| $\lambda$ of 3 comm. | 12 | **88** | 0 |
| **SPECTRE** | | | |
| $\lambda$ of 2 comm. | **93** | 7 | 0 |
| $\lambda$ of 3 comm. | 96 | **4** | 0 |

Figure 5 & Table 4: Conditioning of the generation on the first 3 smallest eigenvectors as an in-painting task using RePaint Lugmayr et al. (2022) (Figure). Number of communities in the graphs generated with spectrum conditioning (Table). Higher values should appear in the bold diagonal.

**Eigenvalues conditioned generation** In this setup, the eigenvectors at time $t-1$ are computed according to Eq. 4, while the eigenvalues are derived from the target ones through the diffusion process of Eq. 2, *i.e.*,

$$\boldsymbol{\lambda}_{t-1} \sim \mathcal{N}\left(\sqrt{\bar{\alpha}_t}\boldsymbol{\lambda}_0, (1-\bar{\alpha}_t)\,\mathbf{I}\right), \qquad \boldsymbol{\Phi}_{t-1} \sim \mathcal{N}\left(\boldsymbol{\Phi}_{t-1}; \boldsymbol{\mu}_\theta\left(\boldsymbol{\Phi}_t, \boldsymbol{\lambda}_t; t\right), \beta_t\mathbf{I}\right). \tag{7}$$

To validate the generation conditioned on the eigenvalues, we generate graphs of the SBM data distribution by fixing the number of nodes as the average number of nodes of graphs containing 2 and 3 communities (70 nodes). We randomly choose one graph with 2 communities and one graph with 3 communities from the test set, and we consider their spectra. We use these to condition the generation of two sets of 100 graphs, for the 2 and 3 communities eigenvalue sequences, respectively. The results reported in Table 4 show that spectral conditioning is able to influence the properties of the generated graphs, while the spectral conditioning provided by SPECTRE fails to do so.

**Eigenvectors conditioned generation** We adopt a similar strategy to condition on a subset of the eigenvectors. In this case, the portion of $k'$ known eigenvectors $\boldsymbol{\Phi}' = [\phi_0, \dots \phi_{k'}]$ at time $t-1$ are derived from the target ones through the forward diffusion process, while the remaining eigenvectors $\overline{\boldsymbol{\Phi}'} = [\phi_{k'+1}, \dots \phi_k]$ and the eigenvalues are computed according to Eq. 4. The three groups of 4 graphs in Figure 5 show the donor graphs (left) from which the first three eigenvectors were computed, the graphs conditioned on the given eigenvectors using GGSD (center) and SPECTRE (right). The color is the 2D color encoding of 2 of the three first eigenvectors manually selected to highlight the different clusters. The colors from the donor graph have then been transported on the generated graphs' corresponding node (same node index). While GGSD is able to preserve the community structure encoded by the given eigenvectors, in SPECTRE this information is completely lost, causing nodes from the same community in the donor graph to randomly spread over different communities of the generated graph. This may be due to the particular generation mechanism of SPECTRE. Specifically, SPECTRE learns a set of reduced orthogonal bases during training, which are left and right-rotated according to a rotation matrix predicted by a PointNetST Segol & Lipman (2019) network based on some input (generated) eigenvalues. This requires an alignment of the graphs to the learned bases, which makes the training more complex. Our approach on the other hand is fully covariant.

## 6 CONCLUSION

We have introduced GGSD, a diffusion-based generative model for graphs where the spectrum of the graph Laplacian is used to retain structural information while reducing the computational complexity. Our approach has a number of advantages, from the ability to directly generate eigenvectors and eigenvalues, to the possibility of naturally encapsulate node feature information as well as conditioning the generation on target spectral properties.

Our model suffers from two main limitations. Firstly, while using low/high frequencies to reconstruct the spectrum is theoretically grounded, in practice it would be interesting to allow the model to select the most informative frequencies for a given dataset. We will explore this possibility in future work. Secondly, while the diffusion model is linear, the bottleneck of our model is the PPGN-based predictor, which has quadratic complexity. In the future, it would be interesting to investigate alternative methods using a sparse representation of the adjacency matrix reconstructed from the Laplacian.

## ACKNOWLEDGMENTS

L.C. and A.B. are supported by the PRIN 2022 project n. 2022AL45R2 (EYE-FI.AI, CUP H53D2300350-0001). G.M. acknowledges financial support from the European Union *NextGenerationEU* in the framework of the iNEST - Interconnected Nord-Est Innovation Ecosystem (iNEST ECS_00000043 – CUP H43C22000540006). In this regard, the views and opinions expressed are solely those of the authors and do not necessarily reflect those of the European Union, nor can the European Union be held responsible for them. A.T.'s work was partially supported by the project "Perturbation problems and asymptotics for elliptic differential equations: variational and potential theoretic method" funded by the program *NextGenerationEU* and by MUR-PRIN, grant 2022SENJZ3.

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

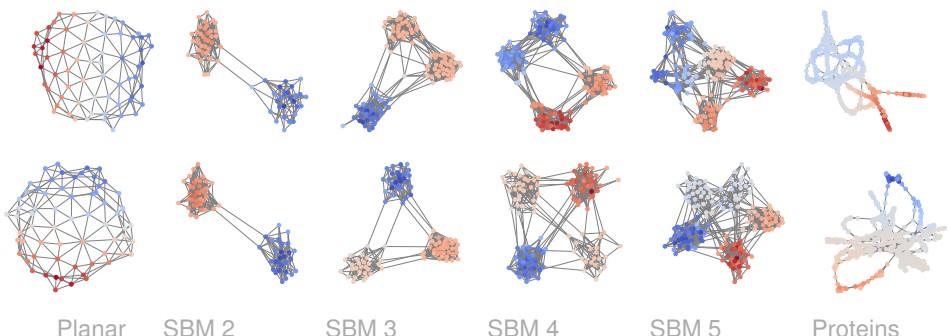

Figure 6: Visual comparison between training set graph samples and generated graph samples produced by GGSD. Each column represents a graph type (Planar, Stochastic Block Model with 2,3,4, and 5 communities, and Proteins). Top row (Original): training set graphs. Bottom row (Generated): graph generated by GGSD.

## A    DATASETS.

We utilize five commonly used datasets for graph generative tasks. Some examples of the graphs contained in these datasets alongside a similar graph generated by GGSD are shown in Figure 6.

**Community-small**: A synthetic dataset consisting of 100 random community graphs, with $[12, 20]$ nodes.

**Stochastic Block Model (SBM)**: A synthetic dataset from Martinkus et al. (2022) comprising 200 Stochastic Block Model graphs, each with a random selection of 2 to 5 communities and 20 to 40 nodes within each community. The probability of edges between communities is set at $0.3$, while the probability of edges within communities is set at $0.05$.

**Planar**: A synthetic dataset from Martinkus et al. (2022) of 200 planar graphs, each containing 64 nodes. Graphs are created through the application of Delaunay triangulation to a randomly and uniformly placed set of points.

**QM9** Ramakrishnan et al. (2014); Ruddigkeit et al. (2012): This real-world dataset comprises 134k organic molecules with a maximum of 9 heavy atoms (carbon, oxygen, nitrogen, and fluorine). By following Simonovsky & Komodakis (2018) we allocate 10k molecules for validation, 10k for testing, and the rest for training.

**Proteins** Dobson & Doig (2003): The dataset encompasses 918 protein graphs, each ranging from 100 to 500 nodes. In this representation, each protein is depicted as a graph, with nodes corresponding to amino acids. Nodes are connected by an edge if they are within a distance of less than 6 Angstroms from each other.

## B    EVALUATION METRICS

### B.1    STATISTICS-BASED

We consider the following key graph statistics: degree distribution (Deg.), clustering coefficient (Clus.), and the occurrence frequency of all 4-node orbits (Orb.). The deviation of these metrics between the generated graphs and the actual ones is measured using the maximum mean discrepancy (MMD) You et al. (2018b). In its initial formulation, the computation of the MMD relied on the Earth Mover's Distance (EMD) and as a result was very slow. For this reason, as suggested in Liao et al. (2019), we use the total variational (TV) Gaussian kernel. This in turn significantly accelerates the evaluation process while maintaining consistency with EMD. In addition to assessing node degree, clustering coefficient, and orbit counts, we also extend our evaluation to include a spectral analysis (Spect.), following Liao et al. (2019). This involves computing the eigenvalues of the normalized graph Laplacian, quantized to approximate a probability density. The spectral

Table 5: Comparison with other graph generative models based on validity, uniqueness, and novelty metrics (the higher the better) on synthetic datasets.

| | **Planar** | | | | **Stochastic Block Model (SBM)** | | | |
| --- | --- | --- | --- | --- | --- | --- | --- | --- |
| | Val.↑ | Uniq.↑ | Nov.↑ | V&U&N ↑ | Val.↑ | Uniq.↑ | Nov.↑ | V&U&N ↑ |
| GraphRNN | 0.00 | - | - | 0.00 | 0.13 | 1.00 | 1.00 | 0.13 |
| GRAN | 0.03 | 1.00 | 1.00 | 0.03 | 0.20 | 1.00 | 1.00 | 0.20 |
| DiGress | 0.70 | 0.98 | 0.95 | **0.65** | 0.13 | 0.98 | 1.00 | 0.13 |
| GSDM | 0.85 | 0.50 | 0.28 | 0.12 | 0.08 | 0.88 | 0.50 | 0.04 |
| GDSS | 0.00 | - | - | 0.00 | 0.01 | 1.00 | 1.00 | 0.01 |
| SPECTRE | 0.14 | 1.00 | 1.00 | 0.14 | 0.51 | 1.00 | 1.00 | **0.51** |
| GGSD | 0.15 | 1.00 | 1.00 | 0.15 | 0.49 | 1.00 | 1.00 | 0.49 |

comparison offers insights into the global properties of the graphs, complementing the local graph statistics emphasized by previous metrics.

## B.2 INTRINSIC-QUALITY-BASED

The *validity* is determined by the ratio of valid molecules to all generated molecules. For molecule graphs (QM9), the validity in molecule generation represents the percentage of chemically valid molecules based on specific domain rules. We measure it using RDKit anitization[1]. For Planar graphs, we use the NetworkX python library based on the left-right planarity test (de Fraysseix & Ossona de Mendez, 2012). For the stochastic block model graphs (SBM), we use a Monte Carlo and greedy heuristic for the inference of the stochastic block model parameters (as implemented by cdlib (Peixoto, 2014)). We consider a graph valid if its probability, assessed by a Wald test (Fahrmeir et al., 2013), of having been generated by the SBM model with inter-edge probability of 0.3 and intra-edge probability of 0.005 is higher than 0.9, and if the number of communities is in the range 2-5 with a number of nodes for each community between 20 and 40.

The *novelty* gauges the percentage of valid graphs that are not sub-graphs of the training set, and vice versa. It checks if the model has successfully learned to generalize to unseen graphs and it considers two graphs identical if they are isomorphic.

The *uniqueness* is defined as the ratio of unique samples to valid and novel samples, measuring the level of variety during sampling. To calculate uniqueness, generated graphs that are sub-graph isomorphic to others are initially removed, and the remaining percentage represents uniqueness. For instance, if a model generates 100 identical graphs, the uniqueness is $1/100 = 1\%$.

The product of these three metrics is referred to as V&U&N (or VUN) and summarizes the ability of the method to generate graphs that are at the same time novel, unique, and valid.

## C  VUN EXPERIMENTS

Table 5 presents the results on the quality of the synthetic graphs generated for the Planar and SBM datasets, analyzing the metrics of validity, uniqueness, novelty, and their combination. As highlighted in Section 5.1, our method achieves 100% uniqueness and novelty on both datasets while showing the second-best score in terms of VUN. Specifically, for SBM, the gap of SPECTRE and our method with respect to other methods is considerable, due to a higher validity score compared to competitors. On the other hand, the validity score of our method on the Planar dataset is lower than the top-performing method (DiGress) of some margin. This behavior is not unexpected and can be explained by the insensibility of the eigenvectors to small local changes of the topology as discussed in Section 5.1. It is worth noticing how GSDM, despite showing the best validity performance on planar graphs, is not able to generate novel graphs, with a novelty score of just 28% in Planar, and 50% in SBM. This does not come as a surprise, since in GSDM the eigenvectors used to reconstruct the final adjacency matrix are uniformly sampled from the training set. This limits the generative power of the method, since most of the information about the graph connectivity is contained in the

---

[1]https://www.rdkit.org/docs/RDKit_Book.html

eigenvectors. As such, the obtained graphs are not actually generated but rather slight modifications of the training set graphs.

## D  MODEL SETTINGS AND IMPLEMENTATION DETAILS

For all datasets except for QM9, we retrained the models using the configurations recommended by the authors. When no recommended hyper-parameters setups or model weights were available, we explored the space of hyper-parameters tuning them according to the ranges mentioned for other datasets. Finally, for our method (GGSD), we use the $k$ largest/smallest eigenvalues of the unnormalized Laplacian. The values of $k$ are experimentally determined as explained in Appendix E. We stress that these are only a fraction of the full set of eigenvectors.

We used the unnormalized Laplacian since it yields an easier graph reconstruction by simple thresholding (as explained in Section 5.2). In order to handle potential scaling issues, we simply normalized the eigenvalues and eigenvectors based on the training data so as to reflect a normal distribution.

For the training of the diffusion model, we split each dataset into 90% train and 10% test, and we train the Spectral Diffusion on the whole dataset for 100k epochs, using early stopping on the reconstruction loss. We performed a grid search on the number of layers between 6, 9 and 12, and selected the best model according to the degree metric computed from the graphs reconstructed directly from the eigenvectors/values and the graphs of the training set. The sampling has been done using DDIM with 200 steps. Moreover, we generate each sample 4 times and keep the one with the lower deviation from orthogonality.

For the training of the Graph Predictor, we used the same splits of the Spectral Diffusion, and trained for 100k epochs. We performed early stopping by comparing the degree distribution of the generated graphs with the training graphs. We used 6 PPGN layers and 3 PPGN layers for the Graph Predictor and the discriminator network respectively, except for QM9 in which also the Graph Predictor is composed of three layers. For QM9, we let the Graph Predictor to generate also edge features, similarly to Martinkus et al. (2022).

For all datasets, following the observations in Appendix E, we train both Spectral Diffusion and Predictor on the 16 smallest and 32 largest eigenpairs and select the final model according to the best average metrics on the validation set.

In order to guarantee the reproducibility of both our model architecture and results, we have made our code accessible on an online public repository [2].

## E  NUMBER OF EIGENVECTORS

To evaluate which part of the spectrum is more relevant and the proper number of eigenvectors to use, we performed an experiment. We trained our model focusing on either the smallest eigenvalues (Smallest) or the largest ones (Largest), while gradually increasing the corresponding number of eigenvectors taken into consideration. The results obtained from two synthetic datasets, Planar and SBM, are shown in Figure 7. Regarding the optimal number of eigenvectors, it appears that too many eigenvectors do not yield the best results, either considering low or high frequencies. Specifically, in the case of the Planar dataset, both smaller and larger eigenvalues exhibit the best performance with 16 eigenvectors. However, for SBM, while the optimal count for lower frequencies remains at 16, for higher frequencies, it increases to 32. These results are not entirely surprising, considering the inherent trade-off between the diffusion model's capability to manage high-dimensional data and the quantity of information (number of eigenvectors) accessible to the Predictor. All in all, it should be noted that the selection of the number of eigenvectors can generally be regarded as a model hyperparameter for optimization, acknowledging its potential dependence on the specific dataset.

Building on the findings outlined above, in our experiments we employed the 32 largest eigenvalues and the 16 smallest eigenvalues - and their corresponding eigenvectors - for all datasets, except for the community-small dataset, for which we utilized the top 8 largest/smallest eigenvalues, and QM9, for which we used the full set of eigenpairs.

---

[2]https://github.com/lcosmo/GGSD

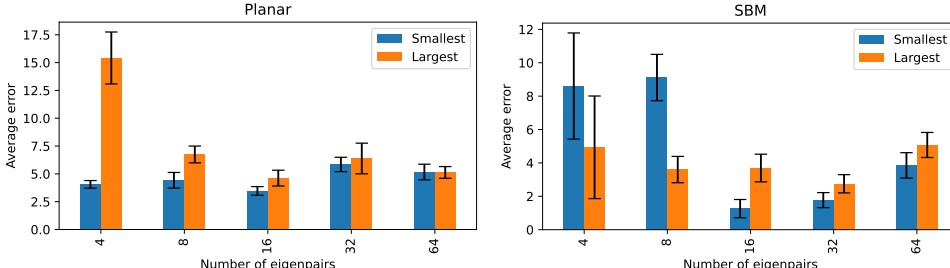

Figure 7: Performance analysis with varying numbers of eigenpairs and different spectrum parts, for the SBM (left) and Planar (right) datasets. The average error represents the mean degradation of all metrics between the generated graphs and the training set. We report both the mean and the standard deviation as error bars on 10 generations of 200 graphs. Specifically, we consider the Degree, Cluster, and Spectral metrics. For each metric, the degradation is computed as the ratio between (1) the MMD value computed between the generated graphs and the training graphs and (2) the MMD computed between the test graphs and the training graphs. A value of 1 indicates that the generated graphs exhibit the same statistical difference wrt the training graphs as the test set graphs do. The ratios computed for each metric are then averaged to get a single value indicating the quality of the generation.

## F  EIGENVECTORS ORTHOGONALITY STUDY

We conducted some experiments to provide both quantitative and qualitative analyses of the orthogonality behavior of the generated eigenvectors. In Figure 8 (left), we show how the generated eigenvectors deviate from forming an orthonormal basis. Here we vary the number of generated eigenvectors between 4 and 12 for Community, and between 4 and 64 for SBM. This choice reflects the fact that the graphs in the Community dataset have between 12 and 20 nodes, while in the case of SBM we observed that using more than 64 eigenvectors appears to lead to a degradation in performance (see Figure 8). As expected, increasing the number of generated eigenvectors introduces greater deviations from orthogonality, which aligns with our findings in Appendix E. In Figure 8 (right), we provide qualitative examples by comparing the generated eigenvectors with those computed from the adjacency matrix predicted by the PPGN. We can observe that in simpler datasets, such as the Community dataset, the eigenvectors are perfectly aligned. In more challenging datasets, such as SBM, while the alignment is not exact, the overall correspondence remains good and significant.

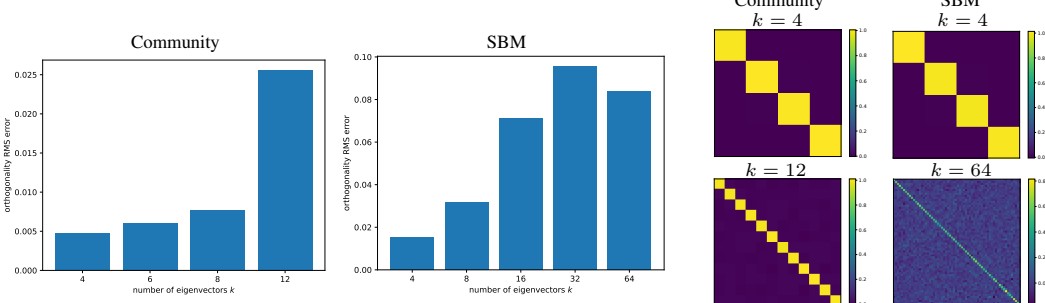

Figure 8: **Left**: the two bar plots show the deviation of the eigenvectors generated by GGSD from an orthonormal basis on two datasets for different numbers of eigenpairs. The deviation from orthogonality is computed as the average root mean squared difference of the inner product of the eigenvectors with the identity matrix, *i.e.*, $RMS(\tilde{\mathbf{\Phi}}_0) = (\frac{1}{k^2}\mathbf{1}(\tilde{\mathbf{\Phi}}_0^\top\tilde{\mathbf{\Phi}}_0 - \mathbf{I})^{\cdot 2}\mathbf{1}^\top)^{\frac{1}{2}}$, with $\cdot^2$ being the elementwise square operator and $k$ the number of eigenvectors. **Right**: Qualitative results showing the inner product of the generated eigenvectors.

| k | 1 | 2 | 3 | 4 | 5 | 6 | 7 | 8 | 9 | 10 | 11 | 12 | 13 | 14 | 15 | 16 |
|---|---|---|---|---|---|---|---|---|---|---|---|---|---|---|---|---|
| SBM | 1.11 | 2.54 | 3.62 | 4.88 | 6.65 | 6.88 | 7.11 | 7.42 | 7.51 | 7.75 | 7.47 | 7.84 | 7.76 | 7.85 | 8.26 | 8.13 |
| Community | 1.09 | 2.97 | 3.41 | 4.18 | 4.71 | 5.46 | 5.77 | 6.13 | 6.46 | 6.79 | 6.87 | 7.54 | - | - | - | - |

Table 6: Average Dirichlet energy computed on 500 generated graphs from the SBM and Community datasets, trained on the lower part of the spectrum (smaller eigenvalues).

The two experiments described above demonstrate that the generated eigenvectors exhibit the smoothness property and are often very similar, sometimes nearly identical, to the eigenvectors computed from the final predicted graph. To further investigate this, we computed the Dirichlet energy of the generated approximate eigenvectors on the final generated graph. We observed a consistent pattern, with the energy increasing as the eigenvalues grow larger. In Table 6, we report the average Dirichlet energy computed from 500 generated graphs in the SBM and Community datasets. We exclude the first eigenvector ($\lambda = 0$) from the analysis, as it does not contribute to the reconstruction of the Laplacian, and the diffusion process is trained only on non-zero eigenvalues.

## G    RUNTIME COMPARISON

Table 7: For each dataset, every method generates 100 graphs. We report the total generation time in seconds.

|  | **Planar** | **SBM** | **Comm.** | **Proteins** | **QM9** |
|---|---|---|---|---|---|
| GraphRNN | 3.30 | 4.19 | 3.16 | **16.87** | — |
| GRAN | 8.58 | 48.43 | 2.90 | 205.47 | — |
| DiGress | 859.64 | 3882.30 | 70.49 | OOM | 68.11 |
| GSDM | 10.71 | 31.51 | 9.74 | 160.09 | — |
| GDSS | 81.10 | 160.88 | 456.30 | 3177.03 | 1.32 |
| SPECTRE | **1.71** | **3.63** | **0.72** | OOM | **0.06** |
| GGSD | 9.51 | 18.63 | 4.41 | 124.07 | 2.50 |

Table 7 shows the total generation time (in seconds) for our method and the baseline methods across the datasets analyzed in this paper. For a fair comparison, we generated 100 graphs for each dataset and method. These experiments were conducted on a computer equipped with an AMD Ryzen 7 3700X processor, 64GB of RAM, and an NVIDIA RTX 3070 8GB graphics card. We achieve a significant speedup compared to DiGress, thus showing that we are able to overcome the computational bottleneck of diffusion-based methods which, unlike GGSD, work on a diffusion space that is quadratic in the number of nodes of the graph. Additionally, our hardware configuration could not register the time for large datasets like Proteins due to their space complexity. While our method outperforms GDSS in terms of speed, it is less efficient than simpler algorithms such as GRAN and GraphRNN.

## H    STABILITY OF THE SPECTRAL DECOMPOSITION OF THE GRAPH    LAPLACIAN

In the spectral graph theory literature, the instability of the Laplacian has become a true-ism. Yet, this claim requires further qualification as several spectral approaches have shown to be robust even under severe deformation Rodolà et al. (2017); Cosmo et al. (2016). In general, random structural perturbation can cause major topological changes which will reflect on the eigenvectors and eigenvalues of the Laplacian quite dramatically, but it strongly depends on the location of the actual perturbation, and it is linked with small gaps in the eigenvalues.

From spectral perturbation theory, we note that under a perturbation $\mathcal{E}$, as long as the eigenvalues are and remain distinct, the eigenvalues of the perturbed Laplacian $\tilde{\mathbf{L}} = \mathbf{L} + \mathcal{E}$ are perturbed by a

quantity

$$\Delta \lambda_i \approx \phi_i^T \mathcal{E} \phi_i, \tag{8}$$

while the eigenvectors are perturbed by $\Delta \Phi \approx \Phi \mathbf{B}$. Here the matrix $\mathbf{B} = (b_{ij})$ is defined as:

$$b_{ij} = \frac{\phi_i^T \mathcal{E} \phi_j}{\lambda_j - \lambda_i}. \tag{9}$$

As a consequence, the mixing can become large even for small perturbations if the gap between the eigenvalues is small, and in general only eigenvectors with close eigenvalues will mix in a significant way. All this being said, this characterizes what happens when we perturb the graph, which is not what is happening here. By recreating the spectrum through a stable diffusion process, the perturbation is in the spectrum, and, in general, small perturbations of the spectrum do not cause major topological changes in the structure (which, as we said, are associated with large spectral variations). Let us say that the eigenvectors are perturbed by a factor of $\Delta \Phi$ and the eigenvalues by a factor of $\Delta \Lambda$, then the reconstructed Laplacian is

$$\tilde{\mathbf{L}} = (\Phi + \Delta \Phi)(\Lambda + \Delta \Lambda)(\Phi + \Delta \Phi)^T = \Phi \Lambda \Phi^T + \underbrace{\Phi \Delta \Lambda \Phi^T + \Phi \Lambda \Delta \Phi^T + \Delta \Phi \Lambda \Phi^T}_{\text{I order error terms}} +$$

$$\underbrace{\Phi \Delta \Lambda \Delta \Phi^T + \Delta \Phi \Delta \Lambda \Phi^T + \Delta \Phi \Lambda \Delta \Phi^T}_{\text{II order error terms}} + \underbrace{\Delta \Phi \Delta \Lambda \Delta \Phi^T}_{\text{III order error term}}, \tag{10}$$

which varies smoothly with noise and does not have elements at the denominator that force the terms to explode. Indeed, we have not observed topological instabilities in any of the datasets considered in this study.

