# OpenReview forum: "Generating  Graphs  via Spectral Diffusion"
_ICLR.cc/2025/Conference — ICLR 2025 Poster_

### Official Review · Reviewer_sbAq · 2024-10-23

**Soundness:** 2
**Presentation:** 3
**Contribution:** 2
**Rating:** 5
**Confidence:** 3

**Summary:**

The paper proposes a graph generative model based on the denoising diffusion probabilistic model (DDPM). The diffusion process is performed on the graph spectral domain.

**Strengths:**

The work combines DDPM and graph spectral decomposition for the proposed generative model. An advantage is that complexity might be reduced as it is necessary to consider the entire graph spectrum. Numerical results demonstrate that the model is effective.

**Weaknesses:**

1. Though combining DDPM and graph spectral decomposition is new to me, there are already works that use DDPM to generate graphs  (e.g., DiGress) and works that use SDE diffusion on graph spectrum (e.g., GSDM). In my opinion, this paper is experimenting with a different combination of diffusion approach and signal domain, which can produce useful results but lacks significant novelty.
2. The model uses the graph eigendecomposition and performs diffusion on the eigenvectors and eigenvalues. However, it is well-known that eigendecomposition (of Laplacian) is highly unstable. I think the authors should theoretically address this issue.
3. Using a part of the spectrum has the advantage of reducing complexity. However, information is lost. What is the balance between these two factors? Is it true that most of the high-frequency components are not important?
4. Is it possible for the diffusion process to generate eigenvectors and eigenvalues that cannot be obtained from the eigendecomposition of any graph?
5. For some datasets (e.g., QM9), the proposed method does not seem to show a clear advantage with a few performance metrics, even though the entire graph spectrum is used. One may gain more insights if the authors can also show the results when a partial graph spectrum is used.
6. How to choose $k$? Is there a principled approach?

**Questions:**

See "Weaknesses".

---

> ### Author Response · Authors · 2024-11-24
>
> We thank the reviewer for the observations and trust that our responses to the points raised help clarify any doubts or concerns.
>
> *"Though combining DDPM and graph spectral decomposition is new to me, there are already works that use DDPM to generate graphs (e.g., DiGress) and works that use SDE diffusion on graph spectrum (e.g., GSDM). In my opinion, this paper is experimenting with a different combination of diffusion approach and signal domain, which can produce useful results but lacks significant novelty."*
> >Our methodology adopts an approach completely different from DiGress and GSDM (which we previously referred to as “FAST” in the original manuscript), and the only part in common is the use of a denoising diffusion framework for generation. In particular, DiGress operates diffusion directly on the nxn adjacency matrix, with the drawback of introducing quadratic computational complexity in the iterative generation process. On the other hand, GSDM proposes to reduce the complexity by performing diffusion on the eigenvalues (not the eigenvectors), and possibly node features, while the eigenvectors for reconstructing the final adjacency matrix are uniformly sampled from the training set. While this strategy allows to speed-up the generation, it actually requires to store the training set as part of the model and limits the generative power of the methods since most of the information about the graph connectivity is contained on the eigenvectors, which are sampled from the training set. As such, the obtained graphs are not actually generated but rather slight modifications of the training set graphs, resulting in a novelty score of just 28% in the Planar dataset, where most of the methods obtain 100% novelty.
> From an application point of view, our method is the only one that allows consistent conditioning on both eigenvectors and eigenvalues.
>
> *"The model uses the graph eigendecomposition and performs diffusion on the eigenvectors and eigenvalues. However, it is well-known that eigendecomposition (of Laplacian) is highly unstable. I think the authors should theoretically address this issue."*
> >It has become some sort of true-ism in spectral graph theory to state that the Laplacian is unstable, but we think that this has to be qualified a bit more as several spectral approaches have shown to be robust even under severe deformation. We have added a further section in the supplementary material (Appendix G) to discuss this from a theoretical point of view in more detail.
>
> *"Is it possible for the diffusion process to generate eigenvectors and eigenvalues that cannot be obtained from the eigendecomposition of any graph?"*
> >Yes, this is actually an intended outcome of our approach. The diffusion process is designed to generate eigenvectors and eigenvalues that may not correspond to the eigendecomposition of any existing graph, allowing for the creation of novel structures.
>
> *"Using a part of the spectrum has the advantage of reducing complexity. However, information is lost. What is the balance between these two factors? Is it true that most of the high-frequency components are not important?"*
>
>
> *"For some datasets (e.g., QM9), the proposed method does not seem to show a clear advantage with a few performance metrics, even though the entire graph spectrum is used. One may gain more insights if the authors can also show the results when a partial graph spectrum is used."*
>
>
> *"How to choose k? Is there a principled approach?"*
>
>
> >We used the entire graph spectrum for QM9 because graphs are small, making the use of a smaller k unnecessary. Nevertheless, we explored the difference between the use of the entire spectrum and a part of it in the community dataset, which is composed of graphs with 12-20 nodes. The results in Table 3 show that, in the case of smaller graphs, it is beneficial to use the full spectrum to reconstruct the adjacency matrix. On the other hand, analyzing the results in Figure 7, we can see that considering a larger number of eigenvectors does not always lead to better performance. In particular, when increasing the eigenvectors, the generative diffusion model struggles to generate significant eigenvectors leading to a decrease in performance. This may be due to the added estimated variance that comes with the increase in dimensionality of the estimation problem. Thus, the trade-off is not just about computational time, but also a classic bias-variance trade-off. For example, we experimentally observed that beyond 16/32 eigenvalues, the noise of the generated eigenvectors increases significantly, which impacts the quality of the generated graphs. This informed our choice of k in the experiments (as also discussed in Appendix D).

---

> > ### Comment · Reviewer_sbAq · 2024-11-25
> >
> > Thank you for the replies. It is nice to see an added appendix on stability. However, my main concern is still about the novelty of the work, which in my opinion that the work is an A+B type of research. Thus, I keep my current assessment of the paper.

---

> ### Author Response · Authors · 2024-11-25
>
> We are sorry the reviewer feels that our work is incremental wrt DiGress and GSDM. As explained above, our work shares little with the mentioned works, except (1) the idea of using denoising diffusion models (or its SDE variant) for generation and (2) approaching the problem from the graph spectrum perspective. These two research areas are too broad to consider any work falling in their intersection just a trivial combination of the two. Moreover, considering the important limitations of previous spectral-based work for graph generation (discussed at length in the paper and in this rebuttal), not only there is still ample room for contribution in this research direction, but efforts (like the present work) to investigate this direction are clearly needed. As such, it's hard to see how our contribution fails to be novel.
>
> In particular, we had to overcome the non-trivial problem of designing a score model architecture capable of handling both eigenvectors and eigenvalues in a principled way while avoiding the quadratic complexity and extracting an adjacency matrix from the Laplacian reconstructed from the truncated basis. Moreover, we show that our method is actually able to preserve the partial spectral characteristics of the eigenvectors/values used for conditioning, a major strength of our method compared with existing literature.  This required the design of a novel approach and wouldn’t have been possible with just minor/incremental improvements of previous methods.
>
> We would be happy to answer to any further comment on this aspect.

---

> > ### Comment · Reviewer_sbAq · 2024-11-27
> >
> > I've gone through the technical part of the revised paper (Sec. 4) again. Frankly, I do not find insightful new ideas, and it is more like an assortment of existing ideas to resolve some technical issues. The main idea of the paper can thus be summarized as "performing DDPM on a subset of the spectrum", which I do not think is sufficiently novel.
> >
> > However, if the authors can theoretically (and rigorously) justify why choosing a subset of the spectrum leads to good performance (in addition to the obvious benefit of having a lower complexity) OR how to theoretically choose the optimal subset, then the new insight will be sufficient for me to raise the score.

---

> > > ### Author Response · Authors · 2024-11-29
> > >
> > > We believe that summarizing our paper as "performing DDPM on a subset of the spectrum" is unfair. Beyond the proposed architecture for the score model, which you may judge not novel enough, ours is the first (and only) method actually generating the graph by generating its eigenvectors and eigenvalues***. Moreover, we performed a vast experimental analysis showcasing the ability of our method to be conditioned on eigenvectors and eigenvalues, maintaining the desired spectral properties on the generated graphs (again, this result is unique to our model). We also experimentally investigated the behaviour of our method with different numbers of eigenvectors, showing the existence of a tradeoff between the number of eigenvectors and the inability of the diffusion model to generate higher dimensional node feature vectors. These are quite exciting results that open up new perspectives on graph generation from spectral information rather than just reaching out to beat the last benchmark. Focusing the judgment on just one section of the whole paper does not do justice to our work and our contribution. For the theoretical bounds on the reconstruction error through the truncated eigenbasis, this is a classical result from spectral theory, i.e.,
> > >
> > > $$L = \sum_{i=1}^n \lambda_i \phi_i \phi_i^\top$$
> > > $$\tilde{L} = \sum_{i=1}^k \lambda_i \phi_i \phi_i^\top, \quad k < n$$
> > > $$\Vert L - \tilde{L} \Vert_F =\left \Vert \sum_{i=k+1}^n \lambda_i \phi_i \phi_i^\top \right \Vert_F $$
> > > $$\Vert L - \tilde{L}\Vert_F = \sqrt{\sum_{i=k+1}^n |\lambda_i|^2}$$
> > >
> > > Here $L$ denotes the Laplacian, $\lambda_i$ and $\phi_i$ are its eigenvalues and eigenvectors, $\tilde{L}$ is the Laplacian reconstructed using $k$ of these eigenpairs, and $\Vert  \cdot \Vert_F$ denotes the Frobenius norm. These equations show that the reconstruction error (in terms of Frobenius norm) is equal to the square root of the sum of the squares of the eigenvalues that are **not** used for the reconstruction of the Laplacian.
> > >
> > > Finally, as mentioned before, the number of eigenvectors to consider cannot be determined theoretically since the main obstacle to keeping more eigenvectors is the ability of the diffusion model to handle too many eigenvectors. We will add a new section in the Appendix with more discussion on this.
> > >
> > > *** *We show that SPECTRE cannot produce graphs with the spectral properties used during conditioning; rather, it uses them just as a conditioning signal without any study on how it influences the generation. GSDM does not generate eigenvectors at all.*

---

> > > > ### Comment · Reviewer_sbAq · 2024-11-30
> > > >
> > > > Let me make my "final" clarification for my assessment. There are already existing diffusion models (DDPM, SDE) and diffusions have also been considered for both the graph and spectral domain. In my opinion, the paper studies a possible combination of the model and the domain, but none are due originally to this work. Therefore, to beef up the technical contribution, the paper needs to provide more theoretical insights on why this combination (of model and domain) leads to better performance. By reading Section 4, I found that most of the model components are adopted from other works, and this is why I commented (earlier) that the section contains "an assortment of existing ideas". Moreover, Section 4 is supposed to be the most important part of the paper, while it fails to convince me that the model is "exciting" as claimed by the authors.
> > > >
> > > > However, I agree that the numerical study of the paper is thorough and extensive. Therefore, I think the paper is at the borderline of being accepted or rejected. My personal preference makes me weight theory more, which explains my overall assessment.

---

> > > > > ### Author Response · Authors · 2024-12-01
> > > > >
> > > > > We thank the reviewer for acknowledging the thorough and extensive experimental study, and we really appreciate the time spent on the discussion. We just have different views on what should be considered novel and do not believe that the research using diffusion models applied to spectral quantities should end with just one work (GSDM) that generates just the eigenvalues and, as such, shows significant limits in generating novel graphs.
> > > > >
> > > > > We just want to remark that ours is not intended to be a theoretical contribution to spectral graph theory, even if we provide theoretical connections to spectral perturbation graph theory to provide an intuitive explanation of the behavior of our method in different datasets (e.g. locally breaking the planarity does not affect much its spectral representation).
> > > > >
> > > > > Finally, we do not think that there is a need to provide any specific explanation of why we perform better than other methods based on spectral quantities beyond what we have just discussed; we are simply the only ones actually generating the eigenvectors. This is just a fact. We have already discussed that GSDM does not generate eigenvectors and that SPECTRE is unable to learn to generate them due to its architecture not being a permutation covariant.

---

### Official Review · Reviewer_mxAE · 2024-11-01

**Soundness:** 3
**Presentation:** 2
**Contribution:** 3
**Rating:** 3
**Confidence:** 4

**Summary:**

An approach using spectral properties for efficient graph diffusion followed by a reconstruction module.

**Strengths:**

1. Spectral information is critical for graph topology, and leveraging it directly for diffusion is an interesting approach to explore.
2. The given ablations are interesting to read and follow.
3. The sampling speed is quick, unlike previous diffusion models.
4. My main concerns while reading the paper are mentioned in the conclusion as limitations, which is clear and closes some of my issues.

**Weaknesses:**

1. Experimentally, some key metrics are missing for synthetic datasets, namely the VUN for the planar and for the SBM dataset. I also have some concerns about the VUN metrics for QM9 since the uniqueness does not seem to be very high while the novelty is outstanding. Both metrics do not overlap totally but reflect the diversity of generation from different perspectives. Would the codes be released later (till my review the given anonymous repo is empty) for checking this technically?

2. The introduction / related work emphasizes a lot on traditional graph generation - which may not be the most critical or related work here. More background for the spectra-based method (SPECTRE is included but it would help if there are discussions with more relevant works) can help to clarify the storyline. Generally, the writing/clarity should be improved. Another branch of method that may be related to GRASP is the 'latent diffusion' for graphs - where people may encode a graph to node features, and, different from GRASP, they may denoise based on those learned features and reconstruct the graph based on it. This method in terms of structure is very similar, and the comparison of using learned graph features, and using spectral information directly is an interesting question to check or mention in the writing.

Will consider increasing the score with the concerns being addressed.

**Questions:**

1. The number of eigenvalues and eigenvectors being used, and using either large or small ones seem to be critical for this method - since SBM, Planar, QM9 use very different settings. Would that require lots of tuning to find the proper one? Explanations about your choice of different graph datasets can be helpful for people not familiar with graph spectra.

2. I fail to understand the information in Table 4. What do the values in the Table represent?

3. Have you considered or experimented using other information besides graph spectrals for diffusion?

---

> ### Author Response · Authors · 2024-11-24
>
> We thank the reviewer for the comments. We hope our answers below clarify all doubts and concerns raised in the review.
>
> *"Experimentally, some key metrics are missing for synthetic datasets, namely the VUN for the planar and for the SBM dataset. I also have some concerns about the VUN metrics for QM9 since the uniqueness does not seem to be very high while the novelty is outstanding. Both metrics do not overlap totally but reflect the diversity of generation from different perspectives. Would the codes be released later (till my review the given anonymous repo is empty) for checking this technically?"*
> >We verified using multiple devices that the code is indeed available/accessible in the repository (https://anonymous.4open.science/r/grasp-D237/), as it was since the submission deadline. We are not sure where the issue may be.
> >
> >We are running new experiments to compute the VUN metrics on the synthetic datasets. We will add a further comment with the results as soon as they are available and we will include them in the revised paper.
> >
> >As for the VUN metrics on QM9, we observe that a few small (not novel) molecules tend to be easily generated, which lowers the uniqueness score of our method. Since the novelty is computed only on unique molecules (i.e., duplicated molecules will be considered just once), this does not affect the novelty score much.
>
> *"The introduction / related work emphasizes a lot on traditional graph generation - which may not be the most critical or related work here. More background for the spectra-based method (SPECTRE is included but it would help if there are discussions with more relevant works) can help to clarify the storyline. Generally, the writing/clarity should be improved. Another branch of method that may be related to GRASP is the 'latent diffusion' for graphs - where people may encode a graph to node features, and, different from GRASP, they may denoise based on those learned features and reconstruct the graph based on it. This method in terms of structure is very similar, and the comparison of using learned graph features, and using spectral information directly is an interesting question to check or mention in the writing."*
> >We revised the manuscript by expanding our discussion about what we believe to be the two most relevant spectrum-based graph generative models (i.e., SPECTRE and GSDM) to better highlight the contribution of our method. We would be happy to discuss more methods if the reviewer has specific inputs to other relevant work.
> >
> >
> >Following your suggestion, we have revised the manuscript by adding a brief discussion on two recent latent diffusion methods for graphs, i.e., “Graphusion: Latent Diffusion for Graph Generation” and “Unifying Generation and Prediction on Graphs with Latent Graph Diffusion”. Indeed, eigenvectors can be seen as a node embedding into a lower dimensional latent space, with eigenvalues bringing in global structure information, allowing to draw a parallelism between our method and these related work.

---

> > ### Author Response · Authors · 2024-11-24
> >
> > *"The number of eigenvalues and eigenvectors being used, and using either large or small ones seem to be critical for this method - since SBM, Planar, QM9 use very different settings. Would that require lots of tuning to find the proper one? Explanations about your choice of different graph datasets can be helpful for people not familiar with graph spectra."*
> > >We have used standard datasets widely used by graph generation methods. To choose the number of eigenvectors/values, we performed a study that is discussed in Appendix D. For most datasets the sweet spot appears to be between 16 and 32 eigenpairs. For datasets made of smaller graphs we use the full spectrum instead.
> >
> > *"I fail to understand the information in Table 4. What do the values in the Table represent?"*
> > >We have clarified the information of Table 4 in the revised manuscript. Specifically, in this experiment we randomly chose one graph with 2 communities and one graph with 3 communities from the test set, and we considered their spectra. We then used these to condition the generation of two sets of 100 graphs, for the 2 and 3 communities eigenvalue sequences, respectively.
> > >
> > >In Table 4 we evaluate the number of communities actually present in the generated graphs. The columns of the table refer to the number of communities in the generated graphs. The rows refer to the number of communities of the graph whose spectrum was used to condition the generation. Therefore, the elements of the table show the number of generated graphs having a specific number of communities given a conditioning spectrum. For example, the first row indicates that 76 out of the 100 graphs whose generation was conditioned on the 2 communities spectrum actually have 2 communities, 19 have 3 communities, and 5 have 4 communities (76+19+5=100).
> >
> > *"Have you considered or experimented using other information besides graph spectrals for diffusion?"*
> > >For this work, we have only considered spectral decomposition for the diffusion process and have not experimented with other types of information.

---

> > > ### Author Response · Authors · 2024-11-28
> > >
> > > As a follow-up to our previous response, we would like to inform you that in the uploaded revised version, we have included the VUN values for SBM and Planar, as requested. Specifically, in the supplementary material, we have added a dedicated section containing the results table, along with a brief discussion of the findings.

---

> > > > ### Comment · Reviewer_mxAE · 2024-11-28
> > > >
> > > > Thanks for your response. It clarified some points but didn’t fully address my concerns.
> > > >
> > > > * Writing: The paper adds more discussion on GSDM but doesn’t clearly explain how your method is different. GSDM seems very similar, and this distinction is necessary to discuss in the paper. Also, diffusion-based methods have advanced a lot (both in terms of formulation and performance) since DiGress, yet there’s no discussion neither results comparison over them. This makes it hard to see where your work fits in the broader context.
> > > >
> > > > * Performance: The results also remain unconvincing. On Planar/SBM, VUN scores might be the most important/well-established metrics, but they are 0.15/0.49, which is not strong enough, known that current diffusion/flow based method reach already 90%+. The results of other works also adds confusion: it shows DiGress VUN as 0.65(Planar dataset)/0.13(SBM dataset), while the original DiGress paper reports 0.75/0.74. For Spectre, the VUN score drops significantly from 0.48 to 0.14 on Planar. Given that at leaset planarity is trivial to verify with existing tools, this discrepancy is hard to justify with implementation. Both Spectre and DiGress have been validated by subsequent work, so their performance should be replicable. It’s surprising that the method struggles with basic graph properties like clustering and planarity, especially since it leverages graph-specific information.
> > > >
> > > > Given these issues, I regret to keep my original rating.

---

> > > > > ### Author Response · Authors · 2024-11-28
> > > > >
> > > > > Thanks for your feedback.
> > > > >
> > > > > **Difference wrt GSDM** GSDM generates just eigenvalues, not eigenvectors. While this is a much simpler task (i.e., just $n$ values for each graph), they need to sample valid eigenvectors from graphs of the training set. As discussed, when reconstructing a graph from its spectral decomposition, most of the information is stored in eigenvectors that GSDM is not generating. From this point of view, it slightly perturbs graphs from the training set rather than generating them. This might be the motivation why they do not report any novelty score in the paper.
> > > > >
> > > > > **Other diffusion models** It would help us improve the paper if you could mention which methods you feel are missing from the analysis.
> > > > >
> > > > > **Replicability of other methods** Note that we did not reimplement any of the methods we compared against. Instead, we run the code provided by the authors of each method with the provided hyperparameters. We are not sure what went wrong with DiGress on planar graphs, the generated graphs are indeed visually good, but planarity is very easy to break. We will mention this discrepancy in the discussion. Note that subsequent works appear to report the results on the synthetic datasets from the original papers, rather than retraining the methods as we did. We would be glad to be pointed toward subsequent works that do not simply report the original results. For SPECTRE, (just a clarification, SPECTRE reports a VUN score of 25% on Planar) we had a hard time training it on Planar. We also got in contact with the authors who confirmed its instability on this dataset.
> > > > >
> > > > > **Limits of our approach** As explained in the text (see the revised Section 5.1 and Appendix C), we actually expect our method (which relies on spectral information as the “graph-specific” information) to have issues capturing hard global graph properties such as planarity. Indeed, similar spectra of (non-)planar graphs can lie on opposite sides of the discrimination boundary, e.g., between planar and non-planar graphs. As such, the addition/removal of an edge can easily break the planarity of the graph without significantly affecting its spectral representation. Consider for example a planar graph containing a subgraph composed of 5 nodes connected by 9 edges. Adding the missing connection (10th) between these 5 nodes will make this subgraph a 5-clique, thus rendering the whole graph non-planar. Yet, this is clearly a local transformation that does not affect the spectrum of the graph significantly (see Appendix G in the revised manuscript).
> > > > >
> > > > > **Performance** We believe our key contribution is to give a new perspective on graph generation that allows conditioning on target spectral properties, rather than simply chasing an epsilon-improvement over existing methods, as it is often the practice in machine learning. This ability to condition the generation on spectral properties is unique to our method, as also validated by the experimental analysis. As far as the performance itself is concerned, we are the best performing method on real world datasets.

---

### Official Review · Reviewer_MC9j · 2024-11-02

**Soundness:** 2
**Presentation:** 3
**Contribution:** 3
**Rating:** 6
**Confidence:** 2

**Summary:**

The authors propose a new algorithm for generating graphs based on spectral decomposition by using diffusion models for the resampling of eigenvectors and eigenvalues, which has been crucial in network analysis.

**Strengths:**

Using the Laplacian spectrum allows us to naturally capture the structural characteristics of the graph and work directly in the node space while avoiding the quadratic complexity bottleneck that limits the applicability of other diffusionbased methods.

**Weaknesses:**

1) The motivation of the method is not clear. Why replacing the GAN module could lead to better results?
2) How to choose non-zero eigenvalues $k$. See also my Q2.
3) Some figures are not clear.

**Questions:**

Q1: From my understanding, the authors replace the GAN module in the pipeline of SPECTRE with the diffusion model. Therefore, it would be better to highlight the difference of the two methods. For example in section 6.4, is there any explanation why GRASP works better in preserving the network community when conditional on the spectral than SPECTRE, since both methods are based on spectral decomposition? In addition, in SPECTRE it was mentioned that conditional on the spectra the performance (section 5.1 therein) can be boosted. How is the performance of GRASP in terms of MMD compared to SPECTRE?

Q2: For methods based on spectral decomposition, it is key to choose the number of non-zero eigenvalues k. I noticed that in SPECTRE relatively smaller k such as 2 and 4 can achieve good performance, while for GRASP large k's are needed. Specifically, for the dataset, QM9 SPECTRE only used k=2, while GRASP used all the eigenvalues.  How is the performance of GRASP compared to that of SPECTRE when the number of non-zero eigenvalues is the same? How will the number of non-zero eigenvalues affect the computational time?

Q3: Could other fast sampling algorithms of diffusion models such as DEIS, DPM-Solver++, UniPC, and so on be leveraged to improve the quality or speed of GRASP?

Q4: Figure 8 is a bit confusing. The first two graphs in the right panel seem to be the same. Some explanations may help the readers to understand why k=12 for the community and k=64 for the SBM are compared.

Q5. For Figure 7, it is unclear what the authors mean by ‘average errors’.  In the caption, it is said that Degree, Cluster, and Spectral metrics are calculated.

---

> ### Author Response · Authors · 2024-11-24
>
> We thank the reviewer for the positive feedback and insightful suggestions, which helped us better frame our work in comparison to other spectral methods.
>
> *"Q1: From my understanding, the authors replace the GAN module in the pipeline of SPECTRE with the diffusion model. Therefore, it would be better to highlight the difference of the two methods. For example in section 6.4, is there any explanation why GRASP works better in preserving the network community when conditional on the spectral than SPECTRE, since both methods are based on spectral decomposition? In addition, in SPECTRE it was mentioned that conditional on the spectra the performance (section 5.1 therein) can be boosted. How is the performance of GRASP in terms of MMD compared to SPECTRE?"*
> >While both our approach and SPECTRE leverage the graph spectrum, there are significant differences between the two, which are not limited to the replacement of the GAN module.
> >
> >SPECTRE focuses on the generation of the adjacency matrix, conditioned on a set of eigenvectors (which may or may not have been generated themselves). Our experiments show that, while this conditioning improves the metrics of the generated graphs, it does not guarantee that these spectral conditioning properties themselves are present in the generated graphs (Figure 3, Figure 5, and Table 4).
> Our method instead directly generates eigenpairs (not an adjacency matrix) using a specifically designed backbone neural network. This is a significant contribution and a major difference from SPECTRE. As a result, our method is capable of generating graphs that respect the given spectral properties (i.e., the eigenvectors of the generated graphs are similar to those used to condition the generation).
> >
> >We have revised the text to make this important distinction clearer.
> >
> >The subpar performance of SPECTRE in preserving the network community structure may be due to its particular generation mechanism. Specifically, SPECTRE learns some reduced orthogonal bases during training, which are left and right-rotated according to a rotation matrix generated by a PointNetST network based on some input (generated) eigenvalues. This requires an alignment of the graphs to the learned bases, which makes the training more complex. Our approach, on the other hand, is fully covariant.
> >
> >We have revised the text to stress this point.
> >
> >The performance of SPECTRE and our method in terms of MMD metrics is reported in Tables 1 & 2. As shown in the tables, our performance is comparable to that of SPECTRE while at the same time better preserving the spectral properties (as discussed above).
>
> *"Q2: For methods based on spectral decomposition, it is key to choose the number of non-zero eigenvalues k. I noticed that in SPECTRE relatively smaller k such as 2 and 4 can achieve good performance, while for GRASP large k's are needed. Specifically, for the dataset, QM9 SPECTRE only used k=2, while GRASP used all the eigenvalues. How is the performance of GRASP compared to that of SPECTRE when the number of non-zero eigenvalues is the same? How will the number of non-zero eigenvalues affect the computational time?"*
> >As discussed above, the two approaches are fundamentally different. While, in principle, SPECTRE can generate a graph even without spectral information, our method directly generates eigenvectors and eigenvalues from which the adjacency matrix can be recovered. As such, we require a larger number of eigenvectors to obtain a good initial reconstruction. However, note that the computational complexity of our model is linear wrt the number of eigenvectors considered (k).
>
> *"Q3: Could other fast sampling algorithms of diffusion models such as DEIS, DPM-Solver++, UniPC, and so on be leveraged to improve the quality or speed of GRASP?"*
> >Indeed our approach is based on DDPM, so any of these faster sampling algorithms could be used, likely improving the speed of GRASP. However, we are somewhat doubtful about their impact on the quality of the generated graphs, which would require further investigation.
>
> *"Q4: Figure 8 is a bit confusing. The first two graphs in the right panel seem to be the same. Some explanations may help the readers to understand why k=12 for the community and k=64 for the SBM are compared."*
> >The graphs in the community dataset have between 12 and 20 nodes, which led us to set k=12 for this dataset. For SBM, values beyond k=64 appear to lead to a degradation in performance, as can be seen from the trend shown in Figure 7. We have revised the text to make this clearer.
>
> *"Q5. For Figure 7, it is unclear what the authors mean by ‘average errors’. In the caption, it is said that Degree, Cluster, and Spectral metrics are calculated."*
> >We have revised the caption of Figure 7 to clarify the meaning of ‘average errors’.

---

### Official Review · Reviewer_sjdg · 2024-11-03

**Soundness:** 4
**Presentation:** 4
**Contribution:** 4
**Rating:** 8
**Confidence:** 4

**Summary:**

This paper considers the problem of realistic graph generation. The
proposed approach focuses on the spectral properties of the generated
graphs, that is, the eigenvectors and eigenvalues. Roughly speaking,
the method learns a diffusion denoiser that acts on the eigenvectors
and eigenvalues jointly, in a way that respects desired equivariance
properties of graph generative models. A key step in reducing the
complexity of this procedure is to restrict the diffusion model to a
small number of eigenvectors, allowing for a linear complexity with
respect to the size of the graph. This reduced representation is
accounted for using a graph neural network.

**Strengths:**

This paper is for the most part well-written and easy to
understand. Indeed, a strength of the proposed approach is that the
architecture is not too different from methods that are already
popular in the literature, so I envision practitioners and other
researchers having an easy time reimplementing this.

The experimental results seem quite good. I was particularly impressed
by the fact that GRASP was able to capture graph statistics well, even
though the method starts with spectral features. The experiments were
carefully selected to demonstrate various aspects of the model's
behavior, which provided insight into the problem beyond just claiming
"SOTA" -- indeed, most of the weaknesses that I was thinking of as I
read Sections 1-5 were addressed directly in Section 6. Bravo!

**Weaknesses:**

I did not find this paper to suffer from any glaring weaknesses, but I
do have one concern about the idea of using a small portion of the
spectrum to reconstruct the entire graph structure. As pointed out,
different parts of the spectrum correspond to different aspects of the
graph structure, that is, local vs. global features. Of course, this
is remarked upon as a limitation by the authors, but I would
appreciate some more discussion on the sorts of graphs that can be
generated in light of this.

Suppose we have a family of graphs that statistically vary in both
their global and local properties, in a way where those properties
(global vs. local) do not have strong correlations. I would imagine
that such a family of graphs could not be captured by merely choosing
to restrict generation to the lowest or highest set of eigenpairs. I
suspect that this concern points to a deeper question about spectral
graph theory than is within the scope of this paper, but I would still
be interested to hear from the authors what sorts of graphs they
expect the proposed method is able to capture.

For instance, SBMs are largely characterized by their global
structure, where the local connections follow an Erdos-Renyi pattern
-- thus, it makes sense to generate such graphs by focusing on the
lower spectrum. On the other hand, expander graphs are known to be
very sparse, while also exhibiting certain properties that are global
in nature -- such as being well-connected in some sense. It would be
helpful to have some experiments to see if such a class of graphs
could be generated by the proposed method.

**Questions:**

I would be interested to hear the authors' response to my main point in the weaknesses section: what sorts of graphs do you think the method, as it is presented in the paper, would have a hard time generating?

---

> ### Author Response · Authors · 2024-11-24
>
> We are glad the reviewer enjoyed reading our work. Below, we respond to the concerns highlighted in the review.
>
> *"I did not find this paper to suffer from any glaring weaknesses, but I do have one concern about the idea of using a small portion of the spectrum to reconstruct the entire graph structure. As pointed out, different parts of the spectrum correspond to different aspects of the graph structure, that is, local vs. global features. Of course, this is remarked upon as a limitation by the authors, but I would appreciate some more discussion on the sorts of graphs that can be generated in light of this.
> Suppose we have a family of graphs that statistically vary in both their global and local properties, in a way where those properties (global vs. local) do not have strong correlations. I would imagine that such a family of graphs could not be captured by merely choosing to restrict generation to the lowest or highest set of eigenpairs. I suspect that this concern points to a deeper question about spectral graph theory than is within the scope of this paper, but I would still be interested to hear from the authors what sorts of graphs they expect the proposed method is able to capture. For instance, SBMs are largely characterized by their global structure, where the local connections follow an Erdos-Renyi pattern -- thus, it makes sense to generate such graphs by focusing on the lower spectrum. On the other hand, expander graphs are known to be very sparse, while also exhibiting certain properties that are global in nature -- such as being well-connected in some sense. It would be helpful to have some experiments to see if such a class of graphs could be generated by the proposed method."*
> >Thank you for the positive feedback on our work. For future work, we are exploring methods for automatically selecting eigenvalues, aiming to dynamically adapt the spectrum used based on the dataset characteristics. Regarding the suggestion about expander graphs, it is currently out of scope for this work, but the idea is indeed interesting, and we will look into it in the future.
>
> *"I would be interested to hear the authors' response to my main point in the weaknesses section: what sorts of graphs do you think the method, as it is presented in the paper, would have a hard time generating?"*
> >Our experimental results suggest that planar graphs appear to be more challenging for our approach. Note that one issue with this dataset is that there is no clear class structure but rather the graphs in the dataset are related by a (hard) global graph property. In this context, similar spectra can lie on opposite sides of the discrimination boundary, e.g., between planar and non-planar graphs. As such, the addition/removal of an edge can easily break the planarity of the graph without significantly affecting its spectral representation. We have revised the text of the manuscript accordingly.
> >
> >Consider for example a planar graph containing a subgraph composed of 5 nodes connected by 9 edges. Adding the missing connection (10th) between these 5 nodes will make this subgraph a 5-clique, thus rendering the whole graph non-planar. Yet, this is clearly a local transformation that does not affect the spectrum of the graph significantly (see Appendix G in the revised manuscript).

---

### Official Review · Reviewer_D9zK · 2024-11-03

**Soundness:** 3
**Presentation:** 3
**Contribution:** 3
**Rating:** 8
**Confidence:** 4

**Summary:**

This article introduces a denoising diffusion model for generating graphs  which will look alike graphs of a training dataset. The parameters of the model are learned in a supervised way from a dataset of graphs, and the output is a Laplacian which can be converted to a graph. The model combines 2 elements in its architecture: a first step to learn the process of spectral diffusion (the diffusion mostly connect the eigenvectors and eigenvalues to the one after 1 time step of diffusion -- the model tries to learn the way back, ie. the reverse diffusion process) and a second step which predicts the graph (which an architecture of GNN, here a PPGN from Maron et al., 2019) to generate the Laplacian (and hence the graph). The idea is that the first step provides a noisy version of the Laplacian matrix (e.g., without orthogonality of the eigenvectors ; or with a reduced number of eigenvectors and eigenvalues) and that the second step helps to recover a correct Laplacian (hence graph). The first step, relying of the general idea of denoising some diffusion model to generate new samples (from the initial work of Sohl-Dickstein et al., 2015 and ho et al., 2020); for graphs, diffusions are operated on the set of eigenvectors and eigenvalues of the Laplacian which are considered (classically) as embedding of the graphs.

The article combines ideas coming from other works (with many points inspired by the structure SPECTRE of Martinkus et al, 2022; or by DiGress from Vignac et al., 20222). Yet the present work comes with some novelties which are well explained. For instance, the diffusion is done on a limited number $k$ of pairs of eigenvectors and eigenvalues, to reduce the memory cost of the model ; the architecture of the neural networks of the 1st step (the learning of the reverse diffusion) is original (with attention heads from eigenvalues to eigenvectors and the converse way as well). The one for the 2nd step relies on existing previous work, yet it is shown in ablative studies that it works well and is needed for good performance.
These new elements are well integrated and explained. An evaluation is carried out on both synthetic datasets, and on real-world datasets of molecular graphs, and compared to 5 or 6 baselines (depending on the experiment), both with some metrics about the relevance of the general structure of the obtained graphs, and some inspections about validity, uniqueness and novelty of the generated graphs seen as molecules. An ablation study is conducted and some additional remarks are made (about orthogonality of the obtained eigenvectors, about the use of the method to generate graphs given a target spectrum for Laplacian, and about runtime (in appendix).

**Strengths:**

This a  quite good article, with a good presentation and interesting ideas.
For me, the strengths of the article are:

* The article is well written, which comprehensive explanations for the method and good presentations of the architecture, the numerical experiments, the ablation study and some inspections of features of the method in 6.3 and 6.4.

* The method builds on previous works, yet it does not feel incremental but more like a thoughtful construction to improve on ESPRIT, DIGRess or other related works.

* The scope is relevant, because it is indeed difficult to build random models of graphs from a limited dataset of graphs to generate new relevant samples. (See however my remark 2 underneath).

* The ablation study is convincing in showing why the two steps are better than only one of the two. Also, there is an interesting discussion and experimental study to see how many pairs of eigenvectors/eigenvalues of $L$ are needed to build a graph and which ones (highest or lowest eigenvalues) are the most appropriate. This element can be explored more in future work and this is an original finding of this article.

**Weaknesses:**

Some weaknesses are:

* The scope is not large, and the work can be seen as somehow incremental. However, these increments are good enough for an article.

* Experimental validations in Section 6 are correct albeit limited due to a small number of datasets. For real-world datasets, it would be good to have examples which are not only related to molecules. Currently, the applications seem too specific.

* The baselines used are ok, yet many results are not computed anew and taken from the published articles. Given the modest number of examples, and the small number of baselines (6 at most), the authors could have tried to re-implement all of them for better control of the reproducibility of the baselines and comparison to the present work.

* Performance is decent, yet not far above (or not above in some cases) the competitors. Some more lines should be devoted to understand why that, and what works better in some other works for some cases.

**Questions:**

Here are some remarks and questions which can help to improve the work:

* 1) the name "GRASP" is already used by several previous work in domains related to the present one: "GRASP: Graph Alignment through Spectral Signatures" of J Hermanns et al., 2021  ; or the GraSP toolbox for Graph Signal Processing (popularized thanks to the A. Ortega's book which uses it). I advise the authors to adopt a different name.

* 2) The 2nd paragraph, p.1 l. 033-042 is not really true, and particularly naive: there are now tons of works to generate random models of graphs with a variety of properties (quoting the Albert and Barabasi model from >20 years ago don't make justice to what is done in the complex network, or network science, community). What is lacking, is most of the time is the precise knowledge of which feature has to be controlled and tuned to a dataset. The present approach which tunes a model to a specific dataset is relevant because of that.

* 3) p.2 l. 071: why should a generative model "assign equal probability to each of these n! adjacency matrices." ? There can be various ways of building probabilities for graphs and why permutation does matter that much ?

* 4) Section 4: I am not certain of the usefulness of that part. This is very basic things which would be incorporated in a part with notations (such a part is missing), and the properties recalled here should be recalled while introducing the work.

* 5) p4, eq. (4): this is the reverse diffusion step, right ?

* 6) Section 5: it would be clearer to have a subsection about step 1 (possibly including 5.1 as a paragraph), then a section about step 2, and maybe a last one about the loss function and how training is done. By the way, even if these questions of training and the two loss functions are inspired by ESPRIT, it would be worth to detail that more (using the space liberated by the removal of section 4).

* 7) Experimental validations in Section 6 is good albeit limited due to a small number of datasets. For real-world datasets, it would be good to have examples which are not only related to molecules.

* 8) Performance themselves is decent, yet, as told above: Some more lines should be devoted to understand why that, and what works better in some other works for some cases.

* 9) in 6.2: why only a threshold of 0.5 is considered ? Given that real-world graphs are often sparse, one could expect a different natural  threshold to obtain desired sparsity.

* 10) In 6.3, are the authors sure of their remark of l. 459 : "...orthonormality..... (indeed, of the eigen-decomposition of any matrix" ? There are matrices with eigendecomposition and non-orthogonal eigenvectors. Your remark holds only for normal matrices.

Edit after revision and discussions: Rating set to 8: accept, good paper

---

> ### Author Response · Authors · 2024-11-24
>
> We thank the reviewer for the constructive remarks. Here below, we answer the questions raised in the review.
>
> *"Experimental validations in Section 6 are correct albeit limited due to a small number of datasets. For real-world datasets, it would be good to have examples which are not only related to molecules. Currently, the applications seem too specific."*
> >We agree that expanding the pool of real-world datasets beyond molecules and proteins would make the work more complete. We plan to include datasets such as Reddit or IMDB to explore broader applications, however, we are unsure if the experiments will be completed by the end of the rebuttal period. Finally, we note that the datasets we considered are those that are commonly used by competing methods, and thus we focus on them to ensure a fair and robust comparison.
>
> *"The baselines used are ok, yet many results are not computed anew and taken from the published articles. Given the modest number of examples, and the small number of baselines (6 at most), the authors could have tried to re-implement all of them for better control of the reproducibility of the baselines and comparison to the present work."*
> >We retrained all the baseline models using the code provided by the authors of the corresponding papers, with the exception of the QM9 dataset, where the results were taken from the original publications (as specified in the caption of Table 2). This is due to the fact that QM9, unlike the other datasets, also contains edge features, with some of the methods not providing the code to train on this dataset. By using the results reported in the literature (for QM9), we were able to still provide a fair comparison.
>
> *"Performance is decent, yet not far above (or not above in some cases) the competitors. Some more lines should be devoted to understand why that, and what works better in some other works for some cases."*
> >We have indeed observed that the performance of our approach appears to be lower on planar graphs. Note that one issue with this dataset is that there is no clear class structure but rather the graphs in the dataset are related by a (hard) global graph property. In this context, similar spectra can lie on opposite sides of the discrimination boundary, e.g., between planar and non-planar graphs. As such, the addition/removal of an edge can easily break the planarity of the graph without significantly affecting its spectral representation. We have revised the text of the manuscript accordingly.
> >
> >Consider for example a planar graph containing a subgraph composed of 5 nodes connected by 9 edges. Adding the missing connection (10th) between these 5 nodes will make this subgraph a 5-clique, thus rendering the whole graph non-planar. Yet, this is clearly a local transformation that does not affect the spectrum of the graph significantly (see Appendix G in the revised manuscript).

---

> > ### Author Response · Authors · 2024-11-24
> >
> > *"The name "GRASP" is already used by several previous work in domains related to the present one: "GRASP: Graph Alignment through Spectral Signatures" of J Hermanns et al., 2021 ; or the GraSP toolbox for Graph Signal Processing (popularized thanks to the A. Ortega's book which uses it). I advise the authors to adopt a different name."*
> > >Thank you for pointing this out. We appreciate your observation and will change the title to “GGSD: Generating Graphs via Spectral Diffusion” to avoid confusion with existing work.
> >
> > *"The 2nd paragraph, p.1 l. 033-042 is not really true, and particularly naive: there are now tons of works to generate random models of graphs with a variety of properties (quoting the Albert and Barabasi model from >20 years ago don't make justice to what is done in the complex network, or network science, community). What is lacking, is most of the time is the precise knowledge of which feature has to be controlled and tuned to a dataset. The present approach which tunes a model to a specific dataset is relevant because of that."*
> > >We apologise for the confusion. In the context of that paragraph, we meant to briefly overview “traditional graph generative model approaches”, as mentioned in the text (a more correct term should have been “seminal”). We recognize that this doesn’t do justice to more recent advancements in this area and we have therefore revised this paragraph accordingly.
> >
> > *"p.2 l. 071: why should a generative model "assign equal probability to each of these n! adjacency matrices." ? There can be various ways of building probabilities for graphs and why permutation does matter that much?"*
> > >As said in the revised text, different permutations correspond to different node orderings for the same graph. We assign equal probability to these permutations because there is no reason to prefer one node order over another. We added the qualifier “possible” to “distinct adjacency matrices” because technically when a graph has symmetries a permutation in the automorphic group of the graph results in the same adjacency matrix. Thus, technically the distribution should be over the n!/|Aut(G)| distinct cosets of the automorphic group Aut(G). However, note that according to the Erdős–Rényi theorem [1], for almost all graphs of sufficiently large size, |Aut(G)|=1.
> > >
> > >[1] Erdős, P., Rényi, A.: Asymmetric graphs. Acta Math. Acad. Sci. Hungar. 14(3), 295–315 (1963)
> >
> > *"Section 4: I am not certain of the usefulness of that part. This is very basic things which would be incorporated in a part with notations (such a part is missing), and the properties recalled here should be recalled while introducing the work."*
> > >We have incorporated a reduced version of Section 4 in “Spectral diffusion” (now Section 4) in the revised manuscript.
> >
> > "*p4, eq. (4): this is the reverse diffusion step, right ?"*
> > >Indeed. We have revised the text just before eq. 4 to make this clear.
> >
> > *"Section 5: it would be clearer to have a subsection about step 1 (possibly including 5.1 as a paragraph), then a section about step 2, and maybe a last one about the loss function and how training is done. By the way, even if these questions of training and the two loss functions are inspired by ESPRIT, it would be worth to detail that more (using the space liberated by the removal of section 4)."*
> > >As suggested, we have restructured Section 5 to include: (1) a very brief overview of the necessary notation (replacing section 4), (2) a subsection for Step 1, and (3) a subsection for Step 2. As for the training/loss details, these are specific to the two parts of the network and thus we believe it is clearer if they remain as part of the two different subsections.
> >
> > *"in 6.2: why only a threshold of 0.5 is considered ? Given that real-world graphs are often sparse, one could expect a different natural threshold to obtain desired sparsity."*
> > >We trained the prediction of the binary adjacency matrix with a binary cross entropy loss. In this context, the choice of a 0.5 threshold corresponds to the Bayes decision rule. Indeed it would be interesting to explore alternative thresholds, however we noticed that the distribution of the resulting value was strongly polarized around 0 and 1. Therefore, we don’t believe a change of threshold would make a significant impact.
> >
> > *"In 6.3, are the authors sure of their remark of l. 459 : "...orthonormality..... (indeed, of the eigen-decomposition of any matrix" ? There are matrices with eigendecomposition and non-orthogonal eigenvectors. Your remark holds only for normal matrices."*
> > >We apologize for the misunderstanding. We meant to write “the eigen-decomposition of any symmetric matrix”. Indeed, it is a well-known fact that symmetric matrices have real eigenvalues and orthogonal eigenvectors. We have revised the text accordingly.

---

> > > ### Comment · Reviewer_D9zK · 2024-11-25
> > >
> > > Reading through the answers provided and the revisions made to the submitted article, I find that this article is now good enough for acceptation. I raise accordingly my rating.

---

### Author Response · Authors · 2024-11-24

We thank all the reviewers for their advice and suggestions to improve the quality of our work. We have uploaded a revised version of the paper, where additions or modifications made in response to the reviewers’ comments are highlighted in red.

In the individual responses to the reviewers, we have indicated where and why the changes were made.

---

### Meta-Review · Area_Chair_WYJ8 · 2024-12-16

**Metareview:**

The paper introduces GRASP (or now called GGSD), a graph generative model leveraging spectral diffusion and eigendecomposition of the graph Laplacian. By sampling eigenvectors and eigenvalues through a denoising process, the model reconstructs the graph while maintaining structural fidelity and reducing complexity via spectrum truncation.

Strengths: integration of spectral methods, linear complexity with respect to the number of nodes, and empirical validation on synthetic and real-world datasets.

Weaknesses: dataset diversity, partial reliance on published baselines, and performance inconsistencies on planar graphs reveal areas for improvement.

**Additional Comments On Reviewer Discussion:**

At the exception of reviewer mxAE, most reviews praised this work for its novelty, efficiency, and empirical validation. The authors' rebuttal addressed most concerns. I recommend acceptance based on the paper's contributions and the authors' responses.

---

### Decision · Program_Chairs · 2025-01-22

Accept (Poster)